# Model Collapse in the Self-Consuming Chain of Diffusion Finetuning: A Novel Perspective from Quantitative Trait Modeling

## Abstract

The success of generative models has reached a unique threshold where their outputs are indistinguishable from real data, leading to the inevitable contamination of future data collection pipelines with synthetic data. While their potential to generate infinite samples initially offers promise for reducing data collection costs and addressing challenges in data-scarce fields, the severe degradation in performance has been observed when iterative loops of training and generation occur—known as "model collapse." This paper explores a practical scenario in which a pretrained text-to-image diffusion model is finetuned using synthetic images generated from a previous iteration, a process we refer to as the "Chain of Diffusion." We first demonstrate the significant degradation in image qualities caused by this iterative process and identify the key factor driving this decline through rigorous empirical investigations. Drawing on an analogy between the Chain of Diffusion and biological evolution, we then introduce a novel theoretical analysis based on quantitative trait modeling. Our theoretical analysis aligns with empirical observations of the generated images in the Chain of Diffusion. Finally, we propose Reusable Diffusion Finetuning (ReDiFine), a simple yet effective strategy inspired by genetic mutations. ReDiFine mitigates model collapse without requiring any hyperparameter tuning, making it a plug-and-play solution for reusable image generation.

## 1 Introduction

*Can state-of-the-art AI models learn from their own outputs and improve themselves?* As generative AI models (e.g., GPT, Diffusion) now churn out uncountable synthetic texts and images, this question piqued curiosity from many researchers in the past couple of years. While some show positive results of self-improving (Huang et al., 2022; Gerstgrasser et al., 2024), most report an undesirable *"model collapse"*—a phenomenon where a model's performance degrades when it goes through multiple cycles of training with the self-generated data (Bertrand et al., 2023; Gillman et al., 2024; Taori & Hashimoto, 2023; Shumailov et al., 2023; Dohmatob et al., 2024a; Fu et al., 2024; Marchi et al., 2024; Martínez et al., 2023b). When large language models (LLMs) are trained with their own outputs, it begins to produce low-quality text that has a lot of repetitions (Dohmatob et al., 2024b), and its linguistic diversity declines rapidly (Guo et al., 2024; Briesch et al., 2023); image models also show quality degradation (Bohacek & Farid, 2023; Martínez et al., 2023a) and loss of diversity (Alemohammad et al., 2023; Hataya et al., 2023).

The goal of this paper is to investigate model collapse in the practical scenario of fine-tuning pretrained text-to-image diffusion models. An end user of text-to-image diffusion models often wants to fine-tune the latest model to generate images with a very specific style (e.g., creating characters in the style of Pokémon). In fact, hundreds of new fine-tuned diffusion models are uploaded regularly on platforms like CivitAI[1], each designed to produce different styles of images. When users scrape the internet to collect images of the style they want, it becomes almost inevitable that synthetic images will be included in their datasets. This is because the number of real images is limited, while synthetic images can be generated in massive quantities and dominate online sources. As a result, users will

---

[1] https://civitai.com/

feel compelled to include more synthetic images in their datasets to keep up with the data demands of these ever-growing, data-hungry models.

To this end, we conduct a thorough investigation into how various hyperparameters commonly used during diffusion fine-tuning (e.g., learning rate, diffusion steps, prompts) impact model collapse. From our extensive empirical analysis, we make a crucial observation: the classifier-free guidance (CFG) scale is the most significant factor that influences the rate of model collapse. Moreover, we observe a fascinating phenomenon that the direction of degradation varies with different CFG values—low CFG results in low-frequency degradation in images, while high CFG leads to high-frequency degradation. The critical role of CFG in determining both the rate and direction of model collapse is a novel insight previously unrecognized in the literature.

To gain a deeper theoretical understanding of this phenomenon, we draw upon the concept of quantitative trait modeling from statistical genetics. Unlike existing studies that attribute model collapse to limited sample sizes and conclude it leads to zero variance (Alemohammad et al., 2023; Bertrand et al., 2023; Shumailov et al., 2023), we propose that the underlying cause is a truncation-based selection process modulated by CFG. This theoretical model accurately describes our experimental observations. Moreover, we show that other theoretical work on model collapse can also be connected to statistical genetics models, such as random genetic drift and its variants (Fisher, 1999; Wright, 1931; Kimura, 1955; Paris et al., 2019). This fresh angle of drawing parallels with statistical genetics offers a new framework to understand the mechanism of model collapse.

Finally, inspired by our empirical results and theoretical insights, we introduce the Reusable Diffusion Finetuning (ReDiFine) method. Our experiments show that selecting an appropriate CFG can significantly slow down model collapse. However, finding the optimal CFG is computationally expensive, requiring numerous iterations of finetuning. To address this, we propose ReDiFine as a plug-and-play solution that achieves a comparable reduction in collapse rate without the need for any hyperparameter tuning. The default ReDiFine setting effectively mitigates model collapse across all four datasets we tested. Moreover, ReDiFine is effective at mitigating model collapse not only in a worst-case scenario of having fully-synthetic dataset, but also in a more practical case, where we have a mix of real and synthetic images the training set. Our contributions can be summarized as follows:

- We performed thorough and extensive empirical investigations into model collapse when we finetune a diffusion model with diffusion-generated images. We tested an exhaustive list of experimental parameters on four datasets (2 digital art & 2 natural images) to identify what affects model collapse. Our analysis reveals that CFG scale is the most influential factor that not only controls the rate of model collapse but also dictates the type of image degradation (Section 3).
- We provide a novel theoretical analysis of model collapse based on quantitative trait modeling that can accurately predict how power spectra of generated images evolve over iterations (Section 4).
- We propose a simple yet effective strategy to mitigate model collapse, named ReDiFine, which combines condition drop finetuning and CFG scheduling. Across all four datasets, ReDiFine successfully generates more "reusable" images, which can be reused to finetune a diffusion model without causing a severe model collapse (Section 5).

## 2 RELATED WORK

The self-consuming training loop and the associated phenomenon known as "model collapse" have become significant areas of study in the past two years (Martínez et al., 2023a;b; Taori & Hashimoto, 2023; Alemohammad et al., 2023; Bohacek & Farid, 2023; Guo et al., 2024; Bertrand et al., 2023; Dohmatob et al., 2024a; Briesch et al., 2023; Gillman et al., 2024; Fu et al., 2024; Marchi et al., 2024). Model collapse, defined as "a degenerative process affecting generations of learned generative models, where generated data end up polluting the training set of the next generation of models" in Shumailov et al. (2023), has been observed in both language and image generative models.

Empirical studies on LLMs (Briesch et al., 2023) reveal that linguistic diversity collapses, especially in high-entropy tasks (Guo et al., 2024), although this can be mitigated with data accumulation (Gerstgrasser et al., 2024). In image generation, several works (Martínez et al., 2023a;b; Alemohammad et al., 2023; Hataya et al., 2023; Bohacek & Farid, 2023; Bertrand et al., 2023) note image degradation when diffusion models are recursively trained with self-generated data. We conduct extensive empirical experiments to reveal the most significant factor causing model collapse in text-to-image

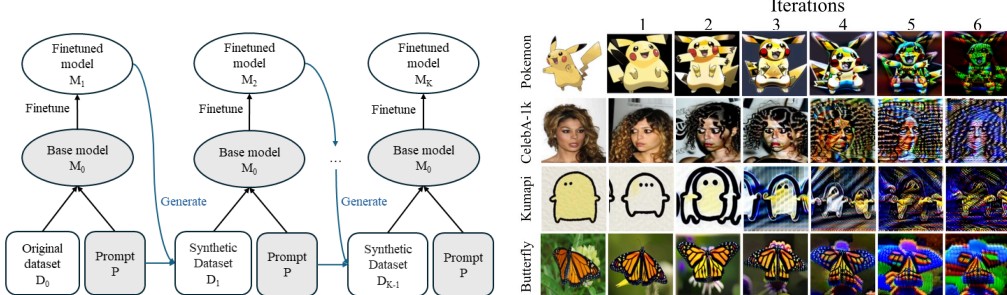

(a) Self-Consuming Chain of Diffusion Finetuning.    (b) Image degradation in the Chain of Diffusion.

Figure 1: (a) **Overall pipeline of the Chain of Diffusion.** Given a pretrained text-to-image diffusion model $M_0$ and a prompt set $P$, a finetuned model $M_k$ is trained using $D_{k-1}$ generated at the previous iteration $k - 1$. Then, $M_k$ generates $D_k$ using the same prompt set $P$, building a fully synthetic loop. Chain of diffusion begins with the original real dataset $D_0$. (b) **Image degradation universally occurs during Chain of Diffusion across four datasets.** As the Chain of Diffusion progresses, the severity of image degradation intensifies, exhibiting universal patterns of highly saturated images containing repetitive high-frequency patterns. This consistently holds across four datasets and 10 scenarios that we comprehensively investigate in Section 3.2.

diffusion models. Our findings reveal that while model collapse is universally observed across various datasets and scenarios, it manifests in three distinct types of image degradation.

Theoretical studies on model collapse have largely focused on the reduction of diversity, typically framed as either decreasing covariance in continuous domains (Alemohammad et al., 2023; Shumailov et al., 2023; Bertrand et al., 2023) or shrinking support in discrete domain (Dohmatob et al., 2024b; Marchi et al., 2024). The finite number of generated samples per iteration has been identified as a key cause of model collapse by several authors (Bertrand et al., 2023; Shumailov et al., 2023; 2024; Dohmatob et al., 2024b; Fu et al., 2024), while others (Alemohammad et al., 2023; Ferbach et al., 2024; Marchi et al., 2024) emphasize that sampling bias reduces the generative model's effective distribution. In contrast, we present a novel theoretical framework based on quantitative trait modeling, shifting the focus from variance reduction to mean drift induced by the selection process. We further demonstrate that many existing theories of model collapse can be understood as extensions of classical results from statistical genetics.

Regarding mitigation strategies for model collapse, existing works echo the importance of incorporating a large proportion of real data throughout the training loop (Alemohammad et al., 2023; Bertrand et al., 2023; Fu et al., 2024; Ferbach et al., 2024) or accumulating additional data (Gerstgrasser et al., 2024). The only mitigation strategy beyond altering the training data composition is proposed by Gillman et al. (2024), who suggests a self-correcting self-consuming loop using an expert model to correct synthetic outputs. While this approach is demonstrated in human motion generation with a physics simulator, having an expert model may not be feasible and too costly for many real-world applications. In our work, we propose an alternative solution through *reusable image generation*, and show that it can mitigate model collapse more effectively than mixing more real data.

## 3   MODEL COLLAPSE IN SELF-CONSUMING CHAIN OF DIFFUSION

### 3.1   PROBLEM SETTING & EXPERIMENTAL SETUP

**Chain of Diffusion.** We begin with formally defining the self-consuming Chain of Diffusion finetuning. Given a pretrained generative model $M_0$, an original training image set $D_0 = \{x_{0,i} | i \in [0, N-1]\}$, and a prompt set $P = \{y_i | i \in [0, N-1]\}$, where $N$ is the number of total images in the dataset, each image $x_{0,i}$ is paired with a corresponding text prompt $y_i$. $M_{k+1}$ is a model finetuned from $M_0$ using the generated image set $D_k = \{x_{k,i} | i \in [0, N-1]\}$ and the prompt set $P$, which simulates a fully synthetic loop (Alemohammad et al., 2023) for finetuning a pretrained generative model. Then, $M_{k+1}$ generates a set of images $D_{k+1}$ for the next iteration using the prompt set $P$:

$$M_{k+1} = \text{Finetune}(M_0, D_k, P), \tag{1}$$

| Candidates | Default | Values |
|---|---|---|
| Dataset size | $1k$ | $0.5k, 2k$ |
| Learning rate | 1e-4 | 1e-3, 1e-5 |
| Finetune | Both | Unet, Text encoder |
| Noise to images | No | Yes |
| Prompt | C | W, S, L |
| CLIP skip | 2 | 1, 3 |
| Diffusion steps | 30 | 100 |
| Epochs | 100 | 50, 200 |
| Mixing real images | 0.0 | 0.5, 0.9 |
| Image per prompt | 1 | 5 |

Figure 2: **Left:** Description of 10 potential factors (excluding CFG) that we examined as candidate sources for model collapse. All experiments were conducted on Pokemon except for the dataset size. For dataset size, we use CelebA since its original dataset is bigger, and we can subsample 500, 1000, and 2000 images. For prompt, we concatenate prompts with different lengths.[2] **Right:** All hyperparameter settings other than changing CFG show a high *collapse rate* greater than 1.0, i.e., FID score increases by $\sim$2x in 6 iterations, indicating severe image degradation. The x-axis represents $FID_1$, quantifying the generation performance at the first iteration and the y-axis represents the *collapse rate* defined in equation 3. For both $FID_1$ and *collapse rate*, lower is better.[3]

$$D_{k+1} = \text{Generate}(M_{k+1}, P). \qquad (2)$$

During the Chain of Diffusion, $M_0$ and $P$ are fixed across all the iterations. To maintain the same size of training dataset, we generate one image per text prompt for all iterations. The overall pipeline of the Chain of Diffusion is shown in Figure 1a.

**Model and datasets.** We use Stable Diffusion v1.5 (Rombach et al., 2022) as the pretrained model $M_0$ and apply LoRA (Hu et al., 2021) to finetune the Stable Diffusion at each iteration. We build our implementation on kohya-ss and perform experiments on four datasets: Pokemon (Pokémon, 2023), Kumapi (Ihelon, 2022), Butterfly (Veeralakrishna, 2020), and CelebA-1k (Liu et al., 2015) to investigate various domains including animation, handwriting, and real pictures. During each iteration, we finetune the pretrained Stable Diffusion $M_0$ for 100 epochs. More implementation details can be found in Appendix A.1 and A.2.

**Evaluation metrics.** We use Frechet Inception Distance (FID) (Heusel et al., 2017) to measure image fidelity. Following Stein et al. (2024), we use DiNOv2 (Oquab et al., 2023) as a feature extractor for FID since it is more consistent with our visual inspection than Inception-V3 network (Szegedy et al., 2016). With a slight abuse of terms, we will still refer to the Frechet distance with DiNOv2 features as the FID score.

In addition to evaluation metrics for generative models presented above, we propose a new metric to quantify the reusability of images generated in the Chain of Diffusion. We define collapse rate as the performance degeneration per iteration in the Chain of Diffusion:

$$\text{collapse rate} = \Delta\text{FID} = \frac{\text{FID}_K - \text{FID}_1}{K - 1}, \qquad (3)$$

where $\text{FID}_k$ stands for the FID between $k$-th iteration set and the original training set. We use FID as a performance metric here, but this can be CLIP, or any other performance metric of interest. Note that a low collapse rate indicates more reusable images since it means that the model does not degrade much during the Chain of Diffusion and we have $K = 6$ in the rest of the paper.

---

[2]C, W, S, and L for Combine, Waifu, Short, and Long, respectively. We concatenate BLIP and Waifu captions as default setting, referred to as Combine. Short and Long prompts are BLIP captions generated with limitations in the lengths of captions. More details can be found in Appendix C.4.

[3]We do not display scenarios that change the training dataset size, such as Img/Prompt and Dataset Size, as varying sizes result in FID scores on different scales. The related results are presented in the Appendix C, where we observe similar image degradation.

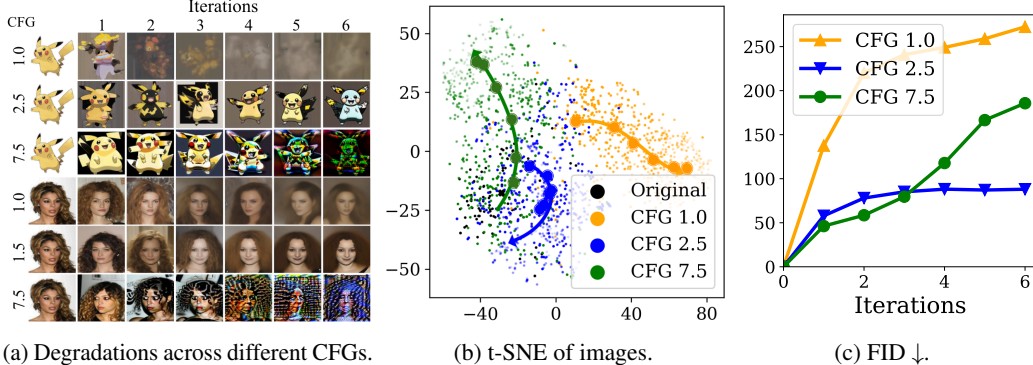

(a) Degradations across different CFGs.  (b) t-SNE of images.  (c) FID ↓.

Figure 3: (a) **Low CFG leads to blurry images and high CFG leads to high-frequency degradation in the Chain of Diffusion.** CFG 2.5 for Pokemon and 1.5 for CelebA exhibit an ideal middle ground where both types of degradation slow down. More images in Appendix B. (b) t-SNE plot visualizes how generated images evolve from the original distribution (black) and shows distinct paths for three CFG scales (Pokemon). Different CFG scales and iterations are differentiated with different colors and transparency. Arrows indicate how the distributions of generated images move for different CFG scales. While CFG 2.5 (blue) stays near the original images (black), high and low CFG scales (1.0 and 7.5) deviate fast, indicating image degradation. (c) Quantitative comparison for FID ↓ (Pokemon). CFG 2.5 achieves the most robust performance. CFG 1.0 degrades from the beginning of the chain while CFG 7.5 begins to degrade in the third iteration, which aligns with the visual inspection in (a).

## 3.2 Model Collapse in the Chain of Diffusion

In this section, we make a series of observations regarding the model collapse behavior in the Chain of Diffusion. We conduct extensive investigations to reveal the most impactful factor in the model collapse and analyze how this factor contributes to the Chain of Diffusion.

**Observation 1: Model collapse is universal in the Chain of Diffusion.** We observe significant image degradation in all four datasets (see Figure 1b) in the Chain of Diffusion. The quality of generated images begins to clearly deteriorate in the third iteration and it drops even more rapidly once the visible degradation emerges, reaching an unrecognizable level in two or three additional iterations. We investigated a variety of different scenarios (summarized in Figure 2) to see if this degradation is an anomaly of specific hyperparameter settings or if it is a ubiquitous phenomenon. We tested various dataset sizes for $D_0$ and $D_k$, increasing the size of synthetic datasets $D_k$ by generating more than one image per prompt, mixing real images from $D_0$ to $D_k$, changing the descriptiveness of prompts, freezing U-Net or text encoder during finetuning, and various other hyperparameters (# sampling steps, # epochs, learning rate, and CLIP skip). We also tested adding a small Gaussian noise in each image in the original set $D_0$ to see if having small random perturbations can improve reusability, and investigated if the degradation happens for larger Stable Diffusion. *In all settings we tested, image degradation was universally present and very fast.* We plot the results for Pokemon on Figure 2 where y-axis is collapse rate and x-axis is the $\text{FID}_1$ (better when closer to the origin). While adding noise to images and mixing 90% real images to every iteration as proposed by Alemohammad et al. (2023); Bertrand et al. (2023); Fu et al. (2024); Ferbach et al. (2024); Gerstgrasser et al. (2024) show the lowest collapse rate, they still exhibit significant degradation as shown in quantitative values (FID score has been doubled in 6 iterations). Visual inspection for images is provided in Appendix C.

**Observation 2: CFG is the most significant factor that impacts the model collapse.** Throughout all our experiments, classifier-free guidance (CFG) had the biggest impact in the speed of model collapse. CFG scale was first introduced in Ho & Salimans (2022) to modulate the strength between unconditional and conditional scores at each diffusion step as follows:

$$\text{Total Score} = \text{Unconditional Score} + \text{CFG} \cdot (\text{Conditional Score} - \text{Unconditional Score}). \quad (4)$$

High CFG means that we emphasize the conditional score for the given prompt more, which pushes the generation to align better with the prompt and often leads to higher-fidelity images. On the other hand, lower CFG places less weight on the conditional score and provides more diversity in generated

images. For those familiar with temperature sampling (Ackley et al., 1985), CFG plays a similar role as temperature, which adjusts the trade-off between fidelity and diversity.

In Figure 2, we observe that as we increase the CFG scale, the image quality in the first iteration improves (smaller $FID_1$ on x-axis), which is expected from our understanding of CFG. A more surprising part is that this comes at the cost of a worse collapse rate (an increase on the y-axis). Also, when the CFG scale is as high as 7.5 or 10.0, the improvement in $FID_1$ plateaus, and increased CFG worsens both $FID_1$ and collapse rate. Similarly, when the CFG scale is too low—below 2.0—the improvement in collapse rate plateaus and both $FID_1$ and collapse rate begin to increase. There is an optimal region of CFG values (near 2.5, specific to Pokemon), where we achieve low collapse rate while maintaining a good quality in the first iteration as well. Moreover, Figure 3c presents FID for Pokemon to demonstrate how different CFG scales affect the performance. CFG scale 2.5 achieves the most robust performance for all metrics for Pokemon. Interestingly, optimal CFG scales differ for different styles: 2.5 for animated or hand-writing datasets (Pokemon and Kumapi) and 1.5 for photo datasets (CelebA and Butterfly) as shown in Appendix B.

**Observation 3: High CFG scales cause high-frequency degradation and low CFG scales cause low-frequency degradation.**   CFG scale does not only affect the speed of model collapse, but also the pattern of model collapse. As shown in Figure 3a, CFG 1.0 makes the images progressively more blurry in the Chain of Diffusion, eventually collapsing to images without any structure, which we refer to low-frequency degradation. On the other hand, for CFG 7.5 how images degrade looks completely different: some features start to be emphasized excessively, repetitive patterns begin to appear, and the overall color distribution becomes saturated. The t-SNE plot in Figure 3b clearly demonstrates that the distribution shift over iterations follows distinct paths for high, low, and medium CFG scales. These patterns were consistent in all four datasets and detailed results can be found in Appendix B. In Section 4, we detail how different patterns of power spectra in high-frequency regions from different CFG scales can be understood using the framework from genetic biology.

**Implications of our observations.**   Our extensive investigations show that a high CFG of 7.5, a common choice to generate visually appealing images, significantly increases collapse rate to achieve slightly better $FID_1$. Sampling for maximizing the perceptual quality was coined as 'sampling bias' in Alemohammad et al. (2023). While they reported a monotonic increase in collapse rate as CFG increased from 1.0 to 2.0, we show that the holistic picture is not entirely monotonic when we look at a wider range of CFG scales from 1.0 to 10.0. It shows an intriguing trade-off between perceptual quality and reusability. This suggests that developers concerning reusability of images can substantially improve future generations by carefully choosing CFG.

## 4   QUANTITATIVE TRAIT MODELS FOR FULLY-SYNTHETIC TRAINING LOOPS

This section introduces a novel perspective to understand the model collapse in generative models by drawing parallels between genetic biology and the Chain of Diffusion. Distinguishable iterations in the Chain of Diffusion—where each iteration is clearly separable with no duplicated individual, and the current iteration originates from the previous one—mirror genetic processes involving successive iterations of parents and offspring. By applying quantitative trait modeling from statistical genetics, we provide a framework to describe how images evolve across iterations in the Chain of Diffusion.

We begin by introducing quantitative trait modeling and its underlying mathematical assumptions. Leveraging these assumptions, we derive a theorem showing that the mean trait value exhibits linear drift, while the variance stabilizes over time. We then show that this model can successfully capture the three key behaviors observed in Section 3: high-frequency degradation, low-frequency degradation, and optimal CFG scale performance. This suggests that different CFG scales in the Chain of Diffusion can be viewed as varying selection strategies, with power spectra corresponding to quantitative traits. We use the term "iteration" instead of "generation" for genetic generation to avoid confusion with image generation.

### 4.1   QUANTITATIVE TRAIT MODELING

Quantitative trait modeling in statistical genetics explores the evolution of quantitative phenotypes (e.g., height, weight, or color), which are decided by multiple genetic and environmental factors

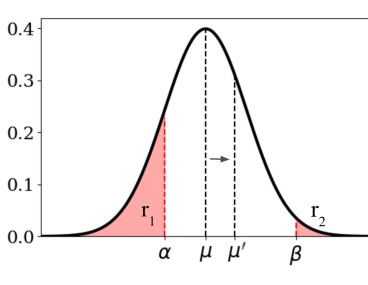

(a) Selection mechanism.

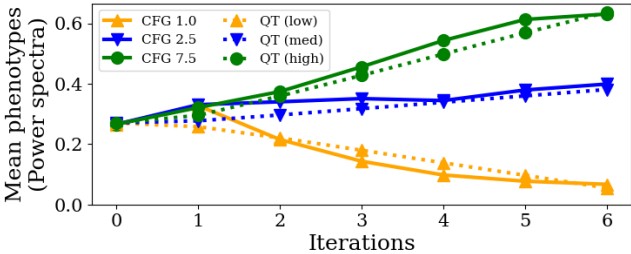

(b) Power spectra of images vs. Mean of phenotypes.

Figure 4: (a) **Selection mechanism with two-sided truncation.** $r_1$ and $r_2$ ratios of samples are truncated from the left and right tails, and the remaining $r = 1 - r_1 - r_2$ ratio of samples is selected. (b) **Power spectra for different CFGs (Pokemon) and quantitative trait modeling results of different selection strategies.** Directional selections with truncation can effectively explain our observations in Section 3: the behaviors of high- and low-frequency degradation and optimal CFG scale. Detailed settings are provided in Appendix D.2.

in complex ways. These phenotypes are typically assumed to follow a Gaussian distribution, with discrete iterations where parents and offspring are distinguishable ($t$- and $t + 1$-th iterations are clearly separable).

Let the distribution of phenotypes at the $t$-th iteration be denoted as $\mathcal{N}(\mu_t, \sigma_{P,t}^2)$ where $\mu_t$ and $\sigma_{P,t}^2$ are the mean and variance of quantitative phenotypes. The phenotypic variance $\sigma_{P,t}^2$ is the sum of the (additive) genetic variance $\sigma_{G,t}^2$ and the environmental variance $\sigma_E^2$, i.e., $\sigma_{P,t}^2 = \sigma_{G,t}^2 + \sigma_E^2$ (Falconer, 1996)[4]. When selection occurs in each iteration, whether natural (e.g., faster animals surviving predators) or artificial (e.g., breeding livestock for higher milk production), it affects the distribution of the effective population that influences the next iteration. We consider directional selection with truncation as shown in Figure 4a, where $r$ ratio of the samples is selected by truncating $r_1$ from the left and $r_2$ from the right side of the distribution. Here, $r + r_1 + r_2 = 1$ and larger phenotype values are preferred when $r_1 > r_2$.

The (narrow-sense) heritability[5], which is defined as the proportion of phenotypic variance attributable to additive genetic factors, can also be represented using the Breeder's Eqn. (Lush, 2013) as:

$$h_t^2 = \frac{\sigma_{G,t}^2}{\sigma_{P,t}^2} = \frac{\sigma_{G,t}^2}{\sigma_{G,t}^2 + \sigma_E^2} = \frac{\mu_{t+1} - \mu_t}{\mu_t' - \mu_t}, \tag{5}$$

where the mean phenotype of the next iteration $\mu_{t+1}$ is represented using the mean phenotype of selected individuals $\mu_t'$, the mean phenotype of the entire population at current iteration $\mu_t$, and heritability $h_t^2$. Additionally, we assume the genetic variance for the next iteration is determined by the variance of selected individuals $\sigma_{G,t+1}^2 = \sigma_{P,t}'^2$. We prove the behaviors of mean and variance of phenotypes under this setting:

**Theorem 1** *Suppose the distributions of phenotypes follow Gaussian distribution and directional selection truncates individuals on both sides with ratios $r_1$ and $r_2$. Then mean of phenotypes asymptotically increases (decreases) by $\frac{c_1 c_2}{\sqrt{1-c_2}} \sigma_E$ per iteration and the variance converges to $\frac{1}{1-c_2} \sigma_E^2$ when $r_1 > r_2$ ($r_2 < r_1$), where $c_1$ and $c_2$ are constants that depend on $r_1$ and $r_2$.*

The proof of Theorem 1 is provided in Appendix D.1.

## 4.2 EXPLAINING THE CHAIN OF DIFFUSION WITH QUANTITATIVE TRAIT MODELING

Just as phenotypes evolve through complex functions of hidden genotypes across multiple iterations in quantitative trait modeling, various continuous features of images evolve similarly during the

---

[4]Genetic variance is composed of additive, dominance, and interaction variance. Here, we only consider additive variance, which is a common assumption in the field.

[5]The heritability is narrow-sense when the genetic variance is restricted to additive variance.

Chain of Diffusion. A key discovery in our work is that the high-frequency power spectra is a crucial phenotype that evolves over iterations and CFG scale acts as different selection mechanisms. These power spectra are computed using 2D Fourier transforms, focusing on frequencies above a certain threshold to capture high-frequency components. Different CFG scales during generation correspond to different selection strategies: a high CFG selects individuals in the right tail of the distribution, favoring more detailed features with reduced diversity, while a low CFG selects from the left tail.

Figure 4b shows the evolution of power spectra in the Chain of Diffusion (in solid line) and the phenotype mean modeled with Eqn. 9 (in dotted lines). The initial mean $\mu_0$ and genetic deviation $\sigma_{G,0}$ in the simulation are set to match those of the original image set at iteration 0. Three different configurations of ratios, $r_1$ and $r_2$, successfully model the power spectra distribution: CFG 7.5 is modeled as selecting from upper 5% of samples, CFG 2.5 selects 50% of the samples slightly favoring higher frequency, and CFG 1.0 results from selecting the lower 30%. We can see that this modeling can capture the evolution of power spectra very accurately over 6 iterations of Chain of Diffuson. Details about power spectra computation and simulation parameters are provided in Appendix D.2.2.

### 4.3 STATISTICAL GENETICS AS A UNIFYING LENS TO MODEL COLLAPSE

Quantitative trait modeling, a tool borrowed from statistical genetics, serves as a solid theoretical framework for interpreting our experimental results. As genetic processes and self-consuming training of generative models share a lot of similarities, we find that other concepts in mathematical genetics also have close ties to existing theoretical work on model collapse. For instance, the traditional Wright-Fisher model (Wright, 1931) describes how traits evolve from one generation to the next in a finite population. Due to randomness in the sampling process, the composition of traits within a finite population drifts over time and eventually collapses to a single phenotype. The Wright-Fisher model and its continuous variants (Tataru et al., 2017) are exactly equivalent to the collapse behavior modeled with simple categorical or Gaussian distributions in Alemohammad et al. (2023); Bertrand et al. (2023); Shumailov et al. (2023; 2024). Given its widespread use in genetics, many extensions of the Wright-Fisher model continue to be an active area of research, such as including different mechanisms of selection (He et al., 2017; Kaj et al., 2024) or mutations (Charlesworth, 2020). Beyond the current work and existing theoretical studies on model collapse, we believe that statistical genetics offers a unified perspective to understand model collapse by reducing the complex dynamics of the self-consuming loop of generative model training to a small number of parameters, from which we can gain valuable insights.

## 5 REUSABLE IMAGE GENERATION WITH REDIFINE

In the previous sections, we have discovered a significant role of CFG in model collapse and that a good choice of CFG can mitigate the collapse effectively. However, the optimal CFG value that minimizes the collapse rate is different for each dataset (e.g., CFG 1.5 for CelebA, CFG 2.5 for Pokemon). There is no efficient way of finding an optimal CFG other than performing a grid search over a wide range of values and iteratively fine-tuning the diffusion model to evaluate the collapse rate for each configuration. In practical scenarios, it is unlikely that end users who are not machine learning experts will go through such a process during finetuning, just to prevent a potential model collapse that can happen in the next generation. This raises the question: *How can we design a finetuning and generation strategy that is user-friendly and can slow down model collapse effectively?*

To address this question, we again draw inspiration from the evolution process in nature. In biology, mutations naturally counteract genetic drift and preserve diversity. Furthermore, selection in nature is often a soft process rather than a hard truncation, as illustrated in Figure 4a. This soft selection allows for the inclusion of outliers, thereby maintaining the overall genetic diversity. We connect these biological inspirations with two strategies: *condition drop finetuning* to include more randomness and *CFG scheduling* during generation to transform hard selection to a softer one[6]. We propose **Re**usable **Di**ffusion **Fine**tuning (ReDiFine) method that incorporates these two ideas and show that it is highly effective at mitigating model collapse without the need for any finetuning.

---

[6]We show a modified quantitative trait modeling result with these two modifications in Appendix D.2 and show that it captures the ReDiFine experimental results effectively. In this section, we focus on showing the image generation results with ReDiFine. We refer the curious readers to the appendix.

Iterations

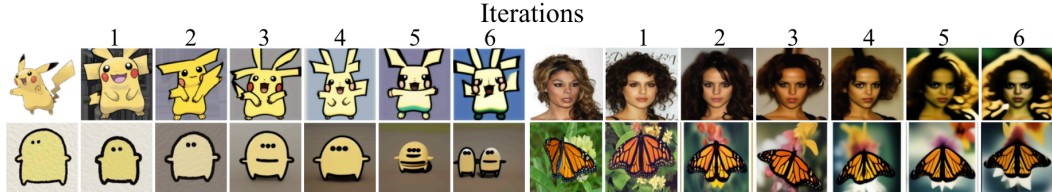

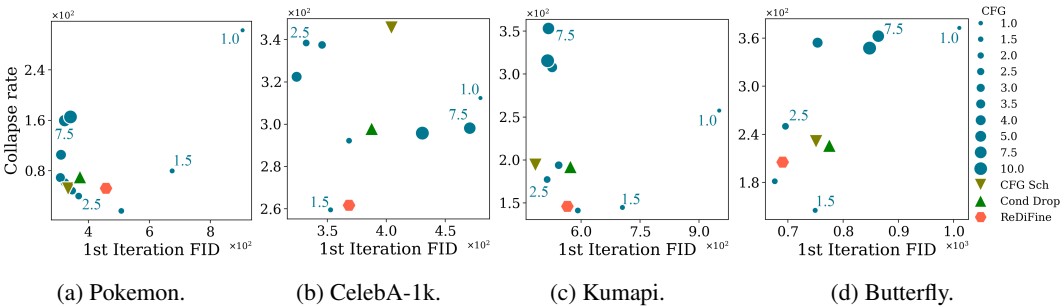

Figure 5: **ReDiFine effectively mitigates image degradation in the Chain of Diffusion.** ReDiFine successfully preserves the characteristics and features without further dataset-specific hyperparameter search. Artifacts observed in high-frequency degradation do not exist for all four datasets.

|  |  |  |  |
|---|---|---|---|
| (a) Pokemon. | (b) CelebA-1k. | (c) Kumapi. | (d) Butterfly. |

Figure 6: **ReDiFine effectively mitigates collapse-FID trade-off across all four datasets**. While the optimal CFG scale varies for different datasets, ReDiFine consistently achieves low collapse rate and FID at the same time (lower is better). Note that the differences in FID are relatively smaller than those in collapse rate, supporting the necessity to evaluate collapse rate in the Chain of Diffusion.

**Condition drop finetuning.** We introduce condition drop finetuning, which randomly drops the text condition during finetuning to update both the conditional and unconditional scores. Although condition drop was suggested in the original CFG paper (Ho & Salimans, 2022), it is not a common practice during finetuning since we can achieve good images without it in the first iteration where model collapse is yet to happen (see Figure 1b). However, the small Diff (= Cond Score − Uncond Score) can accumulate over multiple iterations, leading to a significant gap as the Chain of Diffusion progresses. On the other hand, condition drop finetuning with drop probability $0.2$ preserves the norm of Diff over the iterations (see Figure 38b).

**CFG Scheduling.** We propose gradually reducing the CFG scale during diffusion steps to mitigate the impacts of overemphasizing the conditional score in later stages, which can lead to high-frequency degradation. Specifically, we exponentially decrease the CFG scale $s$ during $T$ diffusion steps as $s = s_0 \times e^{-\alpha \times t/T}$, where $s_0$ is the initial CFG scale and $\alpha$ is the rate of exponential decay. This scheduling approach is consistent with findings by Balaji et al. (2022), which suggest that different diffusion steps contribute uniquely to the generation process. High CFG lets us capture high-level semantic information accurately during the initial diffusion steps while lower CFG in later steps can prevent generating unnecessary high-frequency details.

**ReDeFine Results.** We present the generated images and quantitative metrics for ReDiFine. We use the initial CFG scale $s_0 = 7.5$, diffusion steps $T = 30$, decay rate $\alpha = 2$, and condition drop probability $0.2$ for all of our experiments, except in the robustness comparison and ablation study. While ReDiFine can be further optimized through hyperparameter tuning (e.g., changing condition drop probability), our goal is to demonstrate that ReDiFine robustly mitigates model collapse with these default parameter settings for all datasets. In Figure 5, we can clearly see that ReDiFine significantly improves the image quality at later iterations compared to the baseline (Figure 1b) for all four datasets. In addition to the visual comparison, Figure 6 quantitatively shows the collapse-FID trade-off of ReDiFine. In all four datasets, ReDiFine shows substantially lower collapse rate (y-axis), compared to the baseline case (CFG=7.5). Furthermore, the performance of ReDiFine is close to the optimal Pareto curve spanned by different CFG scales, achieving similar performance as the optimal CFG values across all four datasets, demonstrating its effectiveness as a universal solution for mitigating model collapse. In contrast, using a fixed CFG scale that works well for one dataset often

Figure 7: **Comparison of ReDiFine and baseline (CFG scale 7.5) with training sets mixed with original and synthetic images from the previous iteration.** While mixing real images as much as 90% still results in model collapse for the baseline (as shown in the first row), the ability of ReDiFine to mitigate model collapse is even more highlighted by mixing real images (second row).

fails on others: CFG 2.5 is optimal for Pokemon but performs poorly for CelebA-1k and Butterfly, and CFG 1.5 is optimal for CelebA but performs poorly for Pokemon and Kumapi.

We also compare ReDiFine with real image mixing, which is suggested as a mitigation strategy in several previous papers (Alemohammad et al., 2023; Bertrand et al., 2023; Fu et al., 2024; Ferbach et al., 2024). In Figure 7, we can observe that even infusing the dataset with 90% real images, it shows severe image degradation within six iterations. By visual comparison, we can see that Pikachu images in Figure 5 have less degradation than those with real image mixing. Furthermore, applying ReDiFine to these mixed datasets, as shown in Figure 7, significantly improves image quality compared to the baseline. This demonstrates that ReDiFine is an effective solution for various settings and can be used alongside other mitigation strategies such as real data mixing.

**Ablation study & Further analyses.** We conducted an ablation study to understand the contributions of condition drop finetuning and CFG scheduling to the success of ReDiFine. We plot the results of using only condition drop and CFG scheduling in green triangles on Figure 6. In Pokemon, using only one of those strategies outperforms ReDiFine, but in all other datasets, using only one strategy shows higher collapse rate than ReDiFine. Especially in CelebA, using either one of them showed significantly worse performance than ReDiFine. These results suggest that combining both condition drop and CFG scheduling builds robustness to the method, making ReDiFine effective across all tested datasets. We conducted further analyses on ReDiFine, examining the distribution of latent features, the evolution of the norm of Diff, the power spectra using 2D Fourier transforms, and forensic fingerprints based on prior studies (Corvi et al., 2023a;b). Our analysis shows that ReDiFine effectively preserves the latent distribution and the norms of Diff over six iterations, with forensic fingerprints closely resembling those of the optimal CFG case. Detailed results are provided in Appendix G.

**Reusable image generation as a responsible AI practice.** In this section, we demonstrated that ReDiFine significantly slows the collapse rate without sacrificing image quality, similar to how fair classifiers reduce bias without compromising accuracy (Alghamdi et al., 2022). Adopting ReDiFine is a responsible, environment-conscious practice to prevent polluting the world with images that might look visually appealing, but totally unusable to improve future AI models.

## 6 CONCLUSION

The influx of AI-generated data into the world is inevitable and training sets that consist of synthetic data will be part of the AI development pipeline. In this paper, we empirically and theoretically studied the scenario of finetuning a model with its own generated data, where a gradual degradation called "model collapse" happens. We (1) identify the most impactful factor through comprehensive empirical investigations, (2) develop a novel theoretical perspective inspired by statistical genetics to explain model collapse, and (3) propose an effective mitigation strategy that generates reusable images for future training.

We started this paper with a question: can current AI models learn from their own output and improve themselves? Our paper shows a glimpse that widely-used text-to-image models are not ready to improve from their own creation quite yet. While we presented one solution focused on generating *reusable data*, many open directions remain, such as developing algorithms that can distinguish between real and synthetic data and apply different learning techniques accordingly.

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

APPENDIX

## A EXPERIMENTAL SETUP

### A.1 HYPERPARAMETERS

We finetune Stable Diffusion v1.5 (Rombach et al., 2022) using LoRA (Hu et al., 2021) at each itera-tion, with ft-MSE (StabilityAI, 2022) as a fixed VAE to project images into latent space. Horizontal flip is the only image augmentation applied. Our implementation follows kohya-ss and is built on PyTorch v2.2.2 (Ansel et al., 2024), with torchvision 0.17.2, running on CUDA 12.4 using NVIDIA A-100 and L40S GPUs. All default hyperparameters are listed in Table 1.

Table 1: Default hyperparameters used for the Chain of Diffusion.

| Hyperparameter | Value |
| --- | --- |
| Optimizer | AdamW |
| Learning Rate - Unet | 0.0001 |
| Learning Rate - CLIP | 0.00005 |
| LoRA Weight Scaling | 8 |
| LoRA Rank | 32 |
| Batch Size | 6 |
| Max Epochs | 100 |
| CLIP Skip | 2 |
| Noise Offset | 0.0 |
| Mixed Precision | fp16 |
| Loss Function | MSE |
| Min SNR gamma | 5.0 |
| Max Gradient Norm Clipping | 1.0 |
| Caption Dropout Rate | 0.0 |
| Sampler | Euler A |
| Classifier-Free Guidance Scale | 7.5 |
| Number of Diffusion Steps | 30 |
| Number of Images per Prompt | 1 |

### A.2 DATASETS

We use four image datasets to demonstrate the universal nature of degradation: Pokemon (Pokémon, 2023), Kumapi (Ihelon, 2022), Butterfly (Veeralakrishna, 2020), and CelebA-1k (Liu et al., 2015), covering animation, handwriting, and real images. All images are resized to $512 \times 512$ pixels. Text prompts are generated using BLIP captioner (Li et al., 2022) and Waifu Diffusion v1.4 tagger (Hakurei, 2022). Sample images and prompts can listed in Table 2.

**Pokemon.** The Pokemon dataset (Pokémon, 2023) contains 1008 images indexed by number, with prompts combining BLIP captions (length 50-75 words) and Waifu Diffusion tagger.

**CelebA-1k.** CelebA-1k is a subsample of 1000 images from CelebA (Liu et al., 2015), with BLIP captions (25-50 words).

**Kumapi.** Kumapi (Ihelon, 2022) includes 391 handwriting-style images, with Waifu-generated prompts and manual adjustments.

**Butterfly.** Butterfly (Veeralakrishna, 2020) includes 832 images across 8 species, using BLIP captions (length 50-75 words) and species descriptions.

Table 2: Sample images and prompts for Pokemon, CelebA-1k, Kumapi, and Butterfly datasets.

| **Pokemon** | |
|---|---|
| a green pokemon with red eyes and a leaf on the back of its head and tail, an image of the pokemon character with a red eye and big green tail, all set up to look like it is holding a lea, ultra-detailed, high-definition, high quality, masterpiece, sugimori ken (style), solo, smile, open mouth, simple background, red eyes, white background, standing, full body, pokemon (creature), no humans, fangs, transparent background, claws, Bulbasaur | 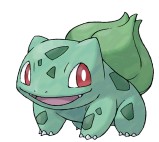 |
| a very cute looking pokemon with a big leaf on its back and a big leaf on its head, a very cute little pokemon character with leaves in the back ground around his chest and hea, white background, ultra-detailed, high-definition, high quality, masterpiece, sugimori ken (style), solo, red eyes, closed mouth, standing, full body, pokemon (creature), no humans, fangs, transparent background, claws, outline, white outline, animal focus, fangs out | 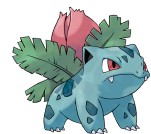 |
| **CelebA-1k** | |
| a woman with brown hair smiling and posing for a picture in front of a mirror and gold and white stripes | 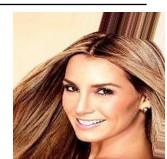 |
| a woman with a very long red hair smiles and laughs on a city street and other people in the background | 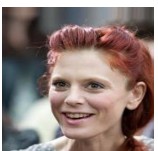 |
| **Kumapi** | |
| solo, simple background, food, donut, grey background, no humans, food focus, still life, Kumapi style | 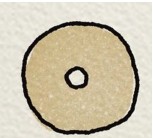 |
| solo, looking at viewer, cute yellow figure, two tiny hands and feet, simple background, black dot eyes, white background, grey background, no humans, Kumapi style | 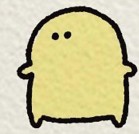 |
| **Butterfly** | |
| a crimson patched longwing butterfly with a red and black stripe on its wings, wings are long, narrow, rounded, black, crossed on fore wing by broad crimson patch, and on hind wing by narrow yellow line | 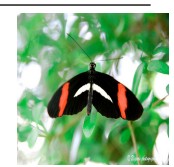 |
| a Common Buckeye butterfly is sitting on a flower in the sun, wings scalloped and rounded except at drawn-out fore wing tip, on hind wing, 1 large eyespot near upper margin and 1 small eyespot below it. Eyespots are black, yellow-rimmed, with iridescent blue and lilac irises, on fore wing, 1 very small near tip and 1 large eyespot in white fore wing bar. | 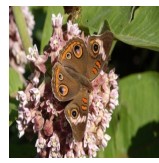 |

## B DIFFERENT CFG SCALES

This section presents images from Chain of Diffusion at various CFG scales for the four datasets. Figure 8, 9, 10, and 11 corresponds to Pokemon, CelebA-1k, Kumapi, and Butterfly, respectively. Each dataset shows results for five CFG scales, covering high, medium, and low values. The optimal medium CFG scale is 2.5 for Pokemon and Kumapi, and 1.5 for CelebA-1k and Butterfly. Lower-than-optimal CFG scales lead to low-frequency degradations, while higher-than-optimal scales result in high-frequency degradations with saturated colors and repetitive patterns. Notably, the optimal CFG for Pokemon and Kumapi causes severe degradation in CelebA-1k and Butterfly.

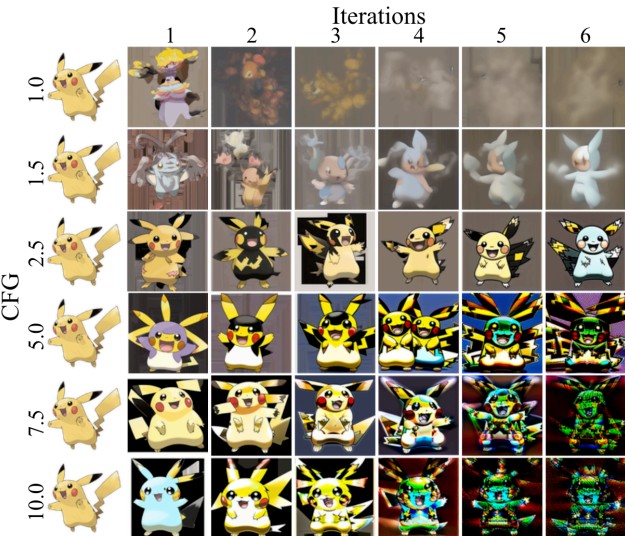

Figure 8: Chain of Diffusion for Pokemon at various CFG scales. The optimal CFG scale is 2.5.

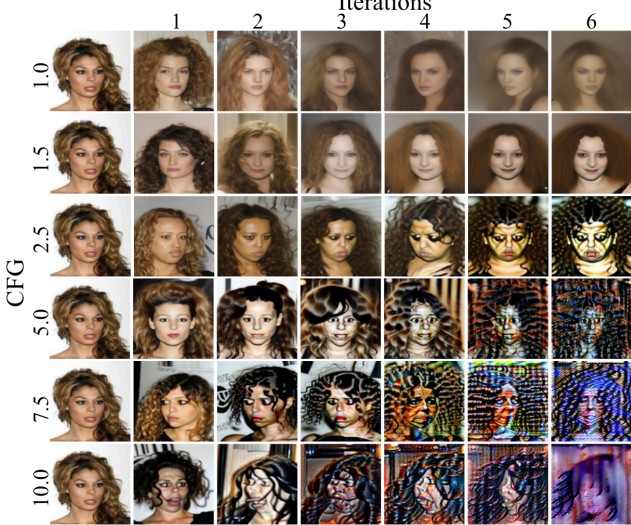

Figure 9: Chain of Diffusion for CelebA-1k at various CFG scales. The optimal CFG scale is 1.5.

## C HYPERPARAMETER INVESTIGATIONS TO UNVEIL THE MOST SIGNIFICANT FACTOR OF DEGRADATION

This section presents experimental results identifying the most significant factors contributing to degradation in the Chain of Diffusion. We systematically vary each hyperparameter from Table **??**,

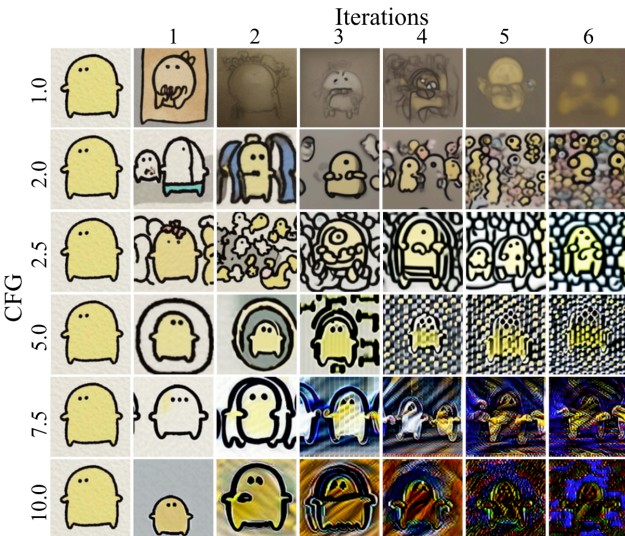

Figure 10: Chain of Diffusion for Kumapi at various CFG scales. The optimal CFG scale is 2.5.

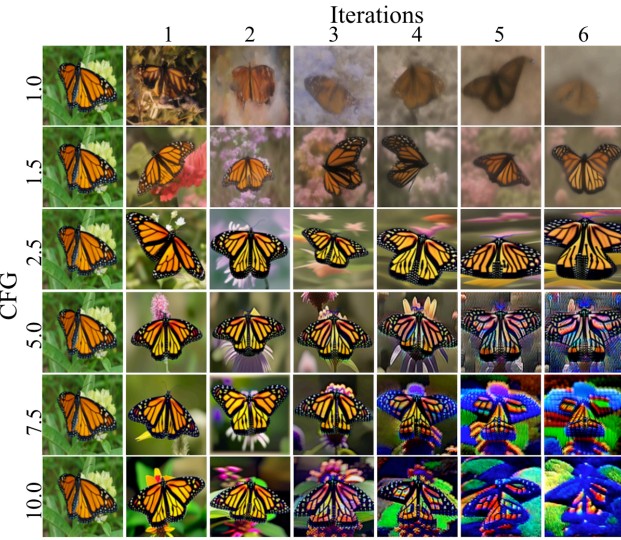

Figure 11: Chain of Diffusion for Butterfly at various CFG scales. The optimal CFG scale is 1.5.

using the default settings from Table 1, to assess their impact on degradation. The CFG scale is fixed at 7.5 for all cases.

## C.1 TRAINING SET SIZE

We used CelebA dataset (Liu et al., 2015) to examine how training set size (both $D_0$ and $D_k$) impacts degradation in the Chain of Diffusion. By subsampling, we adjusted the training set to 100, 250, 500, and 2000 images. Degradation occurs regardless of dataset size, as shown in Figure 12, but appears earlier with smaller sets. By the 6th iteration, images degrade severely for all cases. The number of parameter updates was kept constant across all dataset sizes.

## C.2 NUMBER OF IMAGES PER PROMPT

Generating multiple images per prompt is a simple way to increase training set diversity and can be considered as a solution to mitigate degradation in the Chain of Diffusion. We tested this by

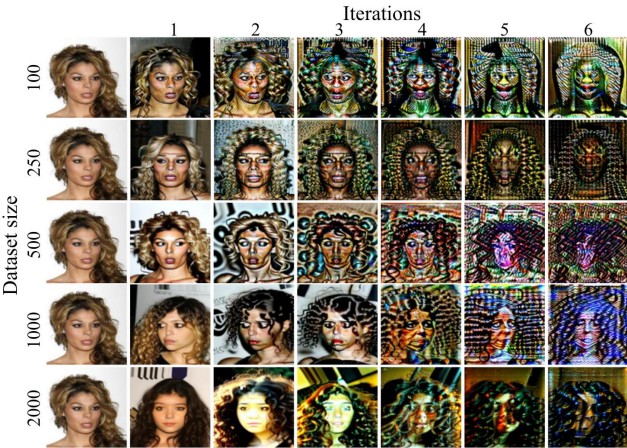

Figure 12: Chain of Diffusion on CelebA dataset with varying training set sizes. Degradation occurs faster with smaller sets, but all result in severe degradation by the 6th iteration.

generating 5 times more images. As shown in Figure 13, while degradation is slightly delayed (by one iteration), it remains unmitigated, and the high computational cost makes this approach impractical.

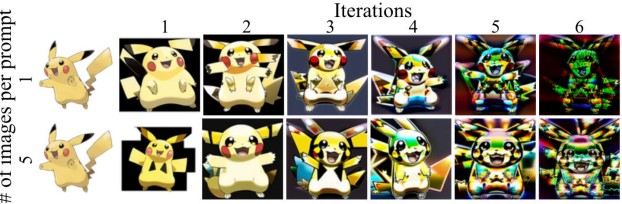

Figure 13: Chain of Diffusion on Pokemon dataset with multiple images generated per prompt. Increasing the training set size in this way does not mitigate degradation.

## C.3 MIXING REAL IMAGES TO SYNTHETIC SETS

Many previous works suggest augmenting the training set with real images to mitigate degradation during iterative training. We investigated whether mixing images from the original training set into the synthetic set at each iteration could alleviate this issue. At each iteration, images are randomly replaced with corresponding real images. Figure 14 and 15 show how degradation in the Chain of Diffusion varies when 50%, 90%, 95%, 98% and 99% of images are replaced for Pokemon and CelebA-1k datasets, respectively. Notably, even 5% synthetic images are sufficient to induce degradation, and 50% replacement rarely slows it down. CelebA-1k dataset appears to be significantly more susceptible to degradation.

## C.4 PROMPT SET

We hypothesized that the descriptiveness of prompts influences degradation in the Chain of Diffusion. We tested various prompt sets for Pokemon and CelebA-1k datasets. For Pokemon dataset, the default prompt set consists of concatenated Waifu and BLIP captions, with BLIP captions ranging from 50 to 75 words. Figure 16 shows how using only Waifu prompts and varying BLIP caption lengths affect degradation. Notably, different styles of high-frequency degradation were observed; shorter prompts reduced repetitive patterns but decreased diversity. In CelebA-1k dataset, varying BLIP caption lengths resulted in similar degradation levels, but prompts that were either insufficiently or excessively descriptive caused the images to deviate from the originals, as shown in Figure 17.

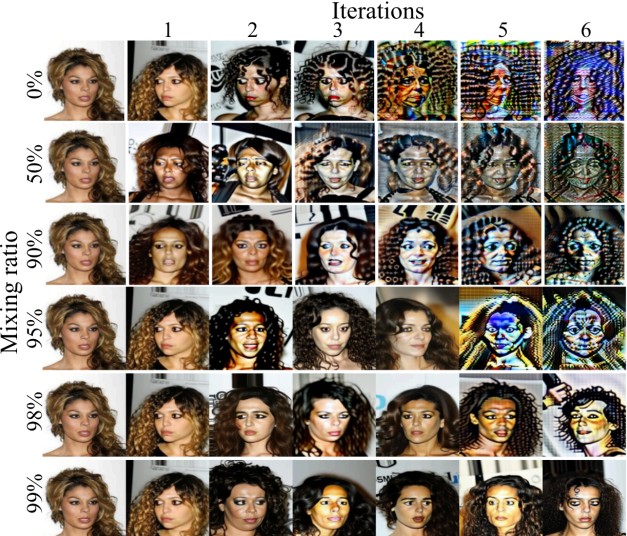

Figure 14: Chain of Diffusion on Pokemon dataset with real images randomly replacing synthetic images at each iteration. A 50% replacement rarely slows degradation, while 10% synthetic images are sufficient to initiate it.

Figure 15: Chain of Diffusion on CelebA-1k dataset with real images randomly replacing synthetic images at each iteration. A 50% replacement rarely slows degradation, while 5% synthetic images are sufficient to initiate it. It suffers from more severe degradation than Pokemon dataset as compared with Figure 14.

## C.5 U-NET AND TEXT-ENCODER

Figure 18 illustrates the Chain of Diffusion with either the U-Net or text encoder finetuned. When the text encoder is not updated (second row), similar degradation occurs. However, the degradation pattern changes when the U-Net is not updated, as the model's ability to generate images remains unchanged. In contrast, updating the text encoder results in a loss of image content preservation.

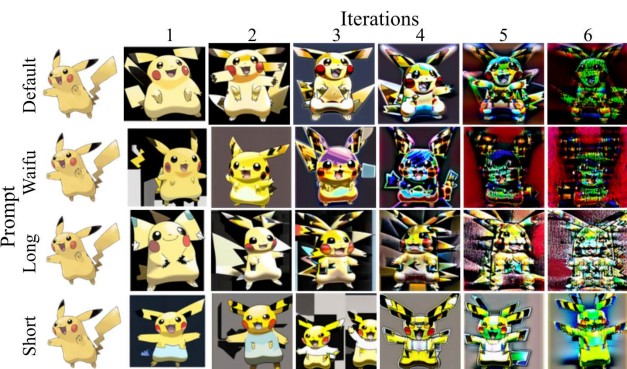

Figure 16: Chain of Diffusion on Pokemon dataset with different prompts. The default prompts are concatenations of BLIP captions (50-75 words) and Waifu captions. We compare the Chain of Diffusion using default captions, Waifu captions, short (less than 25 words) and long (50-75 words) BLIP captions.

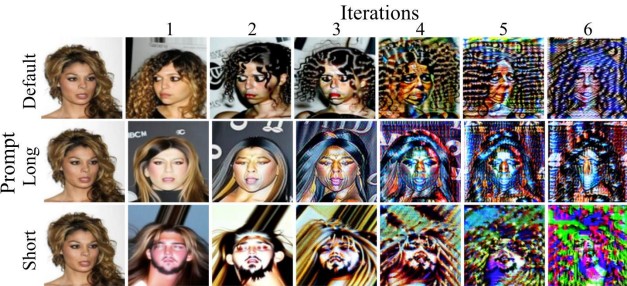

Figure 17: Chain of Diffusion on CelebA-1k dataset with different prompts. The default prompts range from 25 to 50 words. We compare the Chain of Diffusion using longer prompts (over 50 words) and shorter prompts (under 25 words).

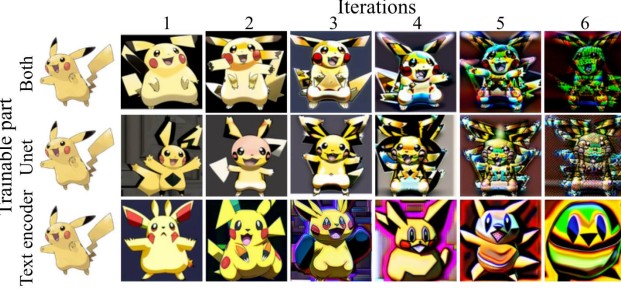

Figure 18: Chain of Diffusion on Pokemon dataset with either the U-Net or text encoder finetuned.

### C.6 NUMBER OF DIFFUSION STEPS

We investigated whether an insufficient number of diffusion steps during generation contributes to degradation. Figure 19 shows that increasing the number of diffusion steps does not enhance the Chain of Diffusion.

### C.7 NUMBER OF EPOCHS

We also examined whether insufficient or excessive training affects our default setting. As shown in Figure 19, images from the initial iterations exhibit similar quality, resulting in comparable degradations. We set the default finetuning to 100 epochs since loss values continue to decrease after 50 epochs.

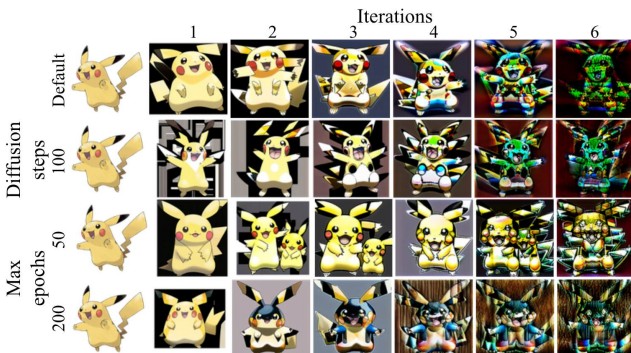

Figure 19: Chain of Diffusion on Pokemon dataset with varying diffusion steps and training epochs. Both increased diffusion steps and differing training epochs fail to mitigate degradation, resulting in similar patterns.

## C.8 LEARNING RATE

Similarly, we assessed finetuning adequacy in Figure 20 by adjusting the learning rates for the U-Net and text encoder by x10 and x0.1. Images from the initial iterations show that the default values are suitable for finetuning. Although the styles are different, degradation consistently occurs across different learning rates.

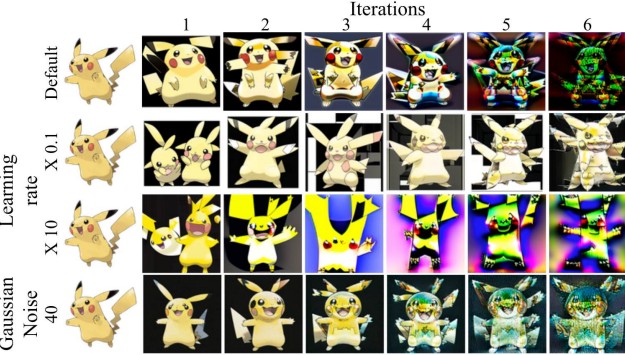

Figure 20: Chain of Diffusion on Pokemon dataset with varying learning rates and added Gaussian noise to the original training set.

## C.9 CLIP SKIP

We investigated the impact of the CLIP skip hyperparameter on degradation. The CLIP skip determines which intermediate feature from the CLIP text encoder is used as the text embedding for conditional generation, with smaller values selecting features closer to the output and larger values selecting those nearer to the input text. As shown in Figure 21, this hyperparameter has minimal effect on degradation patterns.

## C.10 ADDING GAUSSIAN NOISE TO THE ORIGINAL TRAINING SET

We examined whether differences between real and synthetic images contribute to degradation by adding random Gaussian noise to the original training set $D_0$. Figure 20 illustrates that while the characteristics of the original training set have some effect, degradation still occurs. This supports our findings that degradations are universal across real, animation, and handwritten images.

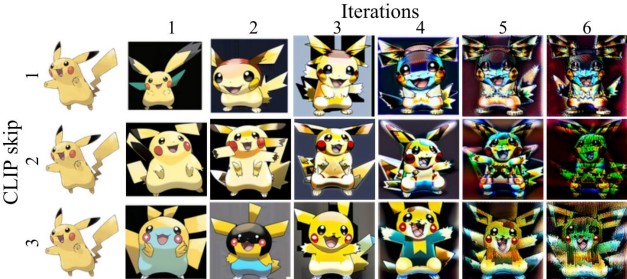

Figure 21: Chain of Diffusion on Pokemon dataset using different CLIP skip hyperparameters. The CLIP skip hyperparameter shows a negligible effect on degradation.

### C.11   STABLE DIFFUSION XL

We investigated image degradation for a different Stable Diffusion model, noting that the optimal hyperparameters for finetuning SDXL using LoRA are not well established. Consequently, we applied the same hyperparameters used for Stable Diffusion v1.5, which may be suboptimal. To manage space complexity, we reduced the batch size to 2 and maintained a resolution of $512 \times 512$, as the first iteration images exhibit impressive quality. Results are presented in Figure 22.

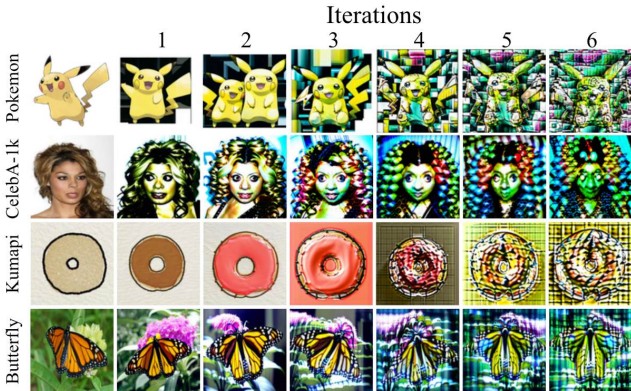

Figure 22: Chain of Diffusion of SDXL on four datasets. Due to insufficient investigation into optimal hyperparameters for finetuning SDXL, our experiments largely rely on those from Stable Diffusion v1.5.

### C.12   ITERATION ACCUMULATION

Iteration accumulation experiments aim to investigate whether concept overfitting and disappearing are major reasons for model collapse. The training set for iteration $t$ is the combination of all previously generated sets, including the original training set. The number of training epochs is controlled accordingly to maintain the total number of updates.

## D   QUANTITATIVE TRAIT MODELING

### D.1   PROOF OF THEOREM 1

*Proof.* The difference between the successive means in quantitative trait modeling is given by:
$$\Delta\mu_t = \mu_{t+1} - \mu_t = h_t^2(\mu_t' - \mu_t). \tag{6}$$
When Gaussian distribution with mean $\mu$ and variance $\sigma^2$ is truncated on both sides with $r_1$ and $r_2$ ratios, the mean and variance of truncated Gaussian distribution are expressed as:
$$\mu_{trun} = \mu - \frac{\varphi(\beta) - \varphi(\alpha)}{\Phi(\beta) - \Phi(\alpha)}\sigma, \tag{7}$$

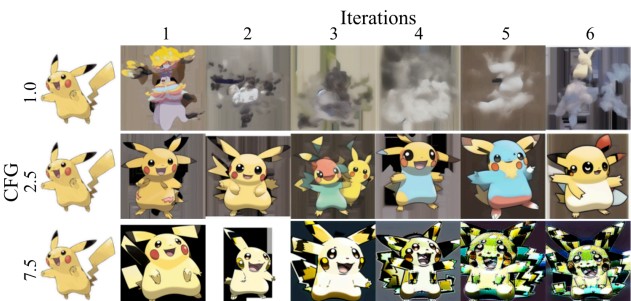

Figure 23: Chain of Diffusion on Pokemon dataset when training set is accumulated from previous iterations. All concepts from previous iterations are preserved for finetuning.

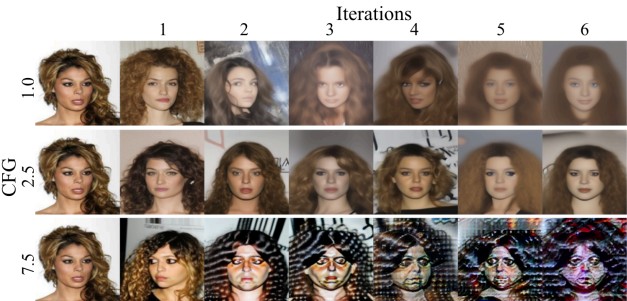

Figure 24: Chain of Diffusion on CelebA-1k dataset when training set is accumulated from previous iterations. All concepts from previous iterations are preserved for finetuning.

$$\sigma^2_{trun} = (1 - \frac{\beta\varphi(\beta) - \alpha\varphi(\alpha)}{\Phi(\beta) - \Phi(\alpha)} - (\frac{\varphi(\beta) - \varphi(\alpha)}{\Phi(\beta) - \Phi(\alpha)})^2)\sigma^2, \tag{8}$$

where $\varphi$ and $\Phi$ are the probability density function (PDF) and cumulative distribution function (CDF) of the standard normal distribution, and $\alpha = \Phi^{-1}(r_1)$ and $\beta = \Phi^{-1}(1 - r_2)$. Accordingly, given $\mu = \mu_t$ and $\sigma^2 = \sigma^2_{P,t}$ with $\mu'_t = \mu_{trun}$ and $\sigma^2_{G,t+1} = \sigma'^2_{P,t} = \sigma^2_{trun}$, we have

$$\Delta\mu_t = h_t^2(\mu'_t - \mu_t) = \frac{\sigma^2_{G,t}}{\sigma^2_{P,t}}c_1\sigma_{P,t} = c_1\frac{\sigma'^2_{P,t-1}}{\sigma^2_{P,t}}\sigma_{P,t} = c_1 c_2 \frac{\sigma^2_{P,t-1}}{\sigma^2_{P,t}}\sigma_{P,t}, \tag{9}$$

where $c_1 = |\frac{\varphi(\beta) - \varphi(\alpha)}{\Phi(\beta) - \Phi(\alpha)}|$ and $c_2 = 1 - \frac{\beta\varphi(\beta) - \alpha\varphi(\alpha)}{\Phi(\beta) - \Phi(\alpha)} - (\frac{\varphi(\beta) - \varphi(\alpha)}{\Phi(\beta) - \Phi(\alpha)})^2$ when $r_1 > r_2$. On the other hand, the mean phenotypes decreases when $r_1 < r_2$ as:

$$\Delta\mu_t = -c_1 c_2 \frac{\sigma^2_{P,t-1}}{\sigma^2_{P,t}}\sigma_{P,t}. \tag{10}$$

Furthermore, the phenotype variance converges over time as:

$$\sigma^2_{P,t} = \sigma^2_{G,t} + \sigma^2_E = \sigma'^2_{P,t-1} + \sigma^2_E = c_2\sigma^2_{P,t-1} + \sigma^2_E. \tag{11}$$

$$\sigma^2_{P,t} - \frac{\sigma^2_E}{1 - c_2} = c_2(\sigma^2_{P,t-1} - \frac{\sigma^2_E}{1 - c_2}) = \cdots = c_2^t(\sigma^2_{P,0} - \frac{\sigma^2_E}{1 - c_2}). \tag{12}$$

As a result, the phenotype variance converges to $\frac{\sigma^2_E}{1 - c_2}$ for $0 < c_2 < 1$ (the variance of truncated distribution is smaller than the variance of the original distribution), and the mean asymptotically increases (decreases) by $\frac{c_1 c_2}{\sqrt{1 - c_2}}\sigma_E$ per iteration when $r_1 > r_2$ ($r_2 > r_1$). $\qquad\square$

## D.2 SIMULATION SETUP

Our simulation aims to demonstrate that our experimental results can be modeled using quantitative trait modeling. We show that the radial sum of power spectra of images is one of the phenotypes explained by our theoretical analysis.

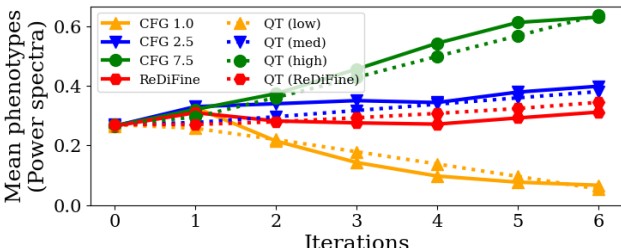

Figure 25: Results comparing the simulation for mutations and power spectra of images generated by ReDiFine (with CFG scale 7.5). Power spectra and simulation are plotted in solid and dotted lines, respectively. Our modifications to heritability and selection process successfully demonstrate changes occur to images by ReDiFine.

### D.2.1 COMPUTING THE RADIAL SUM OF POWER SPECTRA.

The power spectra of images are computed as the square of the magnitude of the 2D Fourier transform. Here, we demonstrate how to compute the radial sum of power spectra in order to use the sum of radial power spectra above a certain threshold. Given a set of images $\{I_i\}$ where $i \in \{1, 2, \cdots, N\}$, the 2D Discrete Fourier Transform (DFT) of an image $I_i$ of size $M \times N$ is computed as:

$$F_i(u, v) = \sum_{x=0}^{M-1} \sum_{y=0}^{N-1} I_i(x, y) e^{-2\pi j (\frac{ux}{M} + \frac{vy}{N})} \tag{13}$$

where $F_i(u, v)$ represents the frequency component at coordinates $(u, v)$. The power spectrum of an image $I_i$ at $(u, v)$ is the square of the magnitude of its Fourier transform $P_i(u, v) = |F_i(u, v)|^2$. We compute the radial sum of the power spectra using a norm of each frequency component $(u, v)$ as $r(u, v) = \sqrt{(\frac{u}{M})^2 + (\frac{v}{N})^2}$. Given the threshold frequency $\tau$, we compute the radial sum of the high-frequency power spectra of an image $I_i$ as

$$S_i = \sum_{u=0}^{M-1} \sum_{v=0}^{N-1} P_i(u, v) \cdot I(r(u, v) > \tau) \tag{14}$$

where $I(\cdot)$ is an indicator function. We compute the sum of high-frequency components because low-frequency components tend to be noisy and use the threshold frequency of $0.02$. Then, the total sum of power spectra for all images can be written as:

$$S = \sum_{i=0}^{N} S_i = \sum_{i=0}^{N} \sum_{u=0}^{M-1} \sum_{v=0}^{N-1} P_i(u, v) \cdot I(r(u, v) > \tau). \tag{15}$$

Practically, we shift the Fourier transform maps so that the frequency norm larger than $\frac{1}{\sqrt{2}}$ is considered to be in the opposite direction. Code for detailed implementation comes from the official code of Corvi et al. (2023a).

### D.2.2 SIMULATION PARAMETERS MATCHING DIFFERENT CFG SCALES.

We simulate different selection strategies using truncation ratios $r_1$ and $r_2$. To model high, medium, and low CFG scales from our experiments, we apply $(0.025, 0.675)$, $(0.5, 0.09)$, and $(0.95, 0.0002)$ for $(r_1, r_2)$, respectively, which correspond to CFG scales of $7.5$, $2.5$, and $1.0$. For initial values in the simulation, we compute the mean $(0.027)$ and standard deviation $(0.056)$ from the original training set (iteration 0), using them as the initial values for mean and genetic standard deviation for our simulation. We set the environmental standard deviation to $0.25$.

### D.2.3 REDIFINE AND MUTATIONS.

ReDiFine combines condition drop finetuning and CFG scheduling, inspired by the mutation mechanism that compensates for distribution shift and maintains genetic distributions in population genetics.

We apply two modifications to our theoretical analysis of Section 4—adding mutation variance to heritability and smoothing truncations—to simulate the effects of ReDiFine in the Chain of Diffusion. Specifically, we add the mutation standard deviation $\sigma_M$ of 0.1 to heritability as:

$$h_t^2 = \frac{\sigma_{G,t}^2}{\sigma_{P,t}^2 + \sigma_M^2} = \frac{\sigma_{G,t}^2}{\sigma_{G,t}^2 + \sigma_E^2 + \sigma_M^2}, \tag{16}$$

representing the randomness added to each iteration due to mutations. This influences the effects of previous iteration to current iteration, which reflect finetuning. Moreover, we apply exponential tails to truncations instead of cut-off thresholds where samples outside the truncation area can be randomly selected with exponential distribution ($e^{-\alpha d(x)}$ where $d(x)$ is a distance to truncation zone and we use $\alpha = 0.1$). This modified selection simulates the effect of CFG scheduling during image generation.

Figure 25 shows the effects of two modifications to our simulation results, and they closely align with the power spectra of images generated by ReDiFine. This demonstrates that the effects of ReDiFine can be understood as interference similar to mutations in population genetics. This suggests further research on model collapse motivated from other fields like biology.

# E    REDIFINE

## E.1    VISUAL INSPECTIONS

Figure 26, 27, 28, and 29 show how robust ReDiFine is to different CFG scales. It successfully mitigates the high-frequency degradations for a wide range of CFG scales.

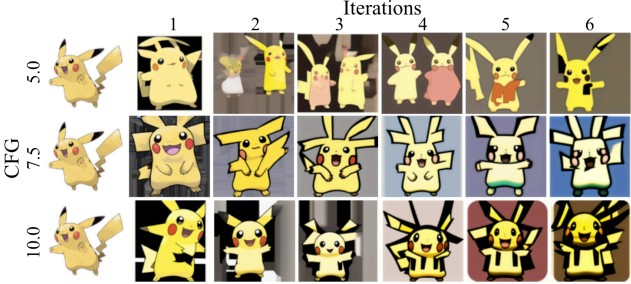

Figure 26: Chain of Diffusion of ReDiFine with different CFG scales on Pokemon dataset. ReDiFine successfully achieves robust image qualities for varying CFG scales.

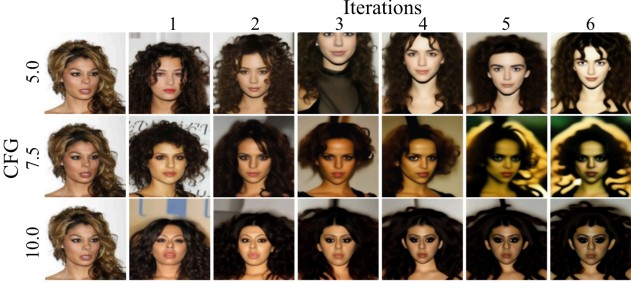

Figure 27: Chain of Diffusion of ReDiFine with different CFG scales on CelebA-1k dataset. ReDiFine successfully achieves robust image qualities for varying CFG scales.

## E.2    MORE ITERATIONS

We conduct additional experiments to compare the baseline with the optimal CFG scale and ReDiFine over extended iterations. As shown in Figure 30, ReDiFine consistently generates images of similar

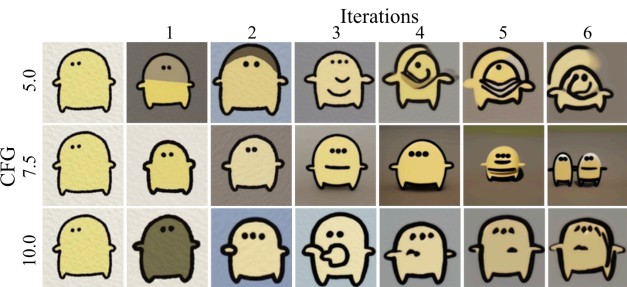

Figure 28: Chain of Diffusion of ReDiFine with different CFG scales on Kumapi dataset. ReDiFine successfully achieves robust image qualities for varying CFG scales.

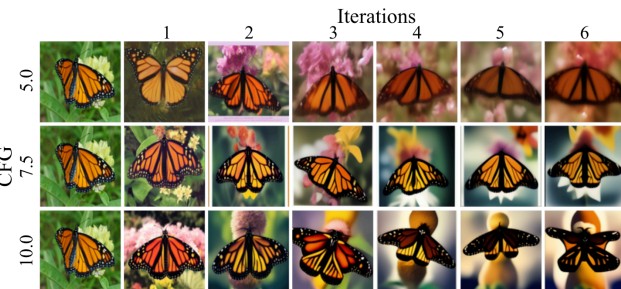

Figure 29: Chain of Diffusion of ReDiFine with different CFG scales on Butterfly dataset. ReDiFine successfully achieves robust image qualities for varying CFG scales.

quality up to 12 iterations, whereas the optimally tuned CFG scale fails to sustain image quality. This decline suggests that repeated hyperparameter searches are necessary to identify suitable CFG scales for subsequent iterations. Such an approach becomes increasingly impractical as the number of iterations grows, highlighting the limitations of relying on the optimal CFG scale to mitigate model collapse.

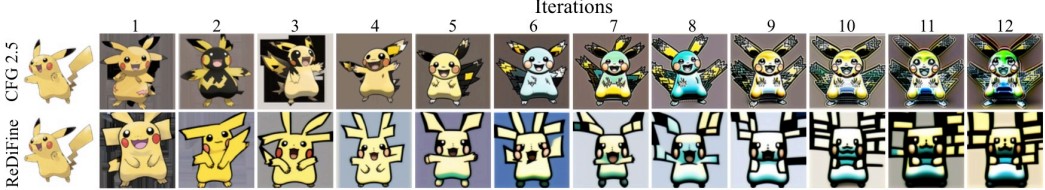

Figure 30: Chain of Diffusion of baseline with the optimal CFG scale and ReDiFine for more iterations. The generation quality of ReDiFine is preserved for additional iterations while optimally found CFG scale 2.5 fails to maintain the image qualities. Similar high-frequency degradation is observed.

### E.3 DATASET ACCUMULATION

We additionally conduct experiments when the training dataset is the accumulation of four datasets (Pokemon, CelebA-1k, Kumapi, and Butterfly) to investigate whether having a broader range of concepts impact model collapse. The accumulated dataset serves as the original training set, while the combined captions are used for image generation at each iteration. The results, presented in Figure 31, show that despite the increased number of images and the inclusion of diverse concepts and domains, model collapse persists at both low and high CFG scales. Moreover, no single CFG scale (e.g., 1.5 or 2.5) can consistently produce high-quality, reusable images across all datasets, highlighting the limitations of relying on an optimal CFG scale for diverse domains. In contrast, ReDiFine leverages the increased conceptual diversity in the original training set, generating more

reliable images across all datasets. To ensure a fair comparison, we control the number of training epochs to maintain consistent updates across experiments.

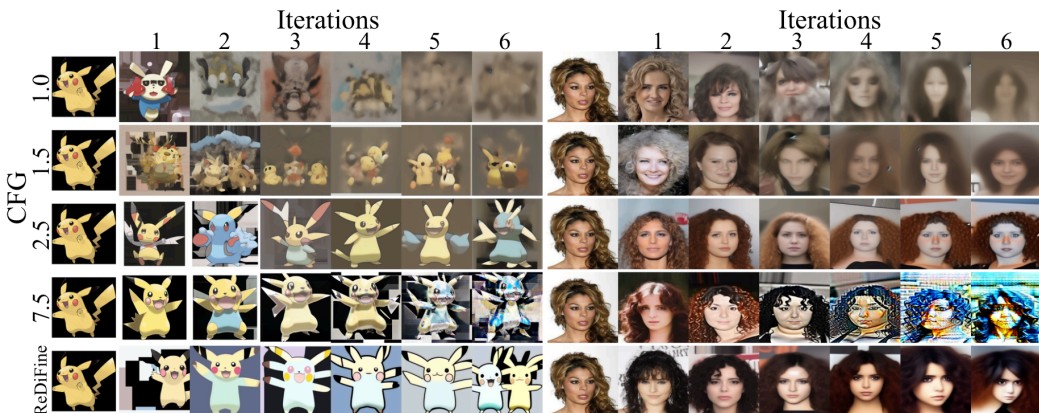

Figure 31: Chain of Diffusion with the accumulation of four datasets (Pokemon, CelebA-1k, Kumapi, and Butterfly) used for finetuning. Model collapse remains evident at low and high CFG scales. A single CFG scale (e.g., 1.5 or 2.5) fails to achieve optimal performance across both the Pokemon and CelebA-1k datasets. In contrast, ReDiFine successfully generates high-quality, reusable images simultaneously. While images for Kumapi and Butterfly datasets are not displayed due to space constraints, they are included in the finetuning process along with the other datasets.

### E.4 ITERATIVE RETRAINING

Some prior works on model degeneration examine scenarios in which a single model is continually trained on synthetic data it has generated. To adapt our Chain of Diffusion framework to this setting, we consider a setup where the same model is finetuned iteratively across multiple iterations. At each iteration, the model generates a fixed number of images using a predefined prompt set, and these generated images are then used to further finetune the model. Figure 32 demonstrates how images degrade under this setting across different CFG scales and ReDiFine. While severe degradation is observed for low and high CFG scales, the optimal CFG scale and ReDiFine are able to mitigate model collapse, generating high-quality images. This indicates that the effect of ReDiFine is maintained even when a single model is continually finetuned.

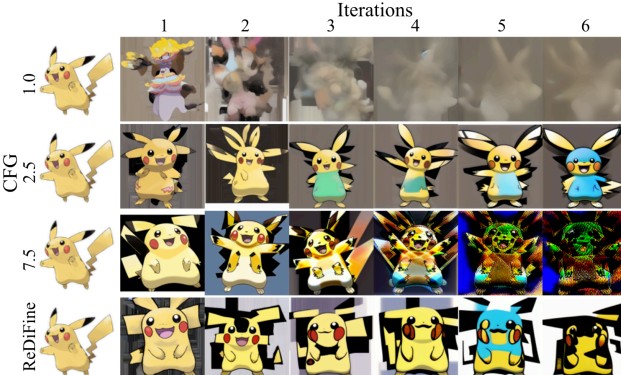

Figure 32: Chain of Diffusion where a single model is continually finetuned across multiple iterations. Model collapse is observed consistently at CFG scales of 1.0 and 7.5, while the baseline with a CFG scale of 2.5 and ReDiFine effectively mitigate model collapse. This demonstrates that model collapse is a universal phenomenon across different settings, and the effect of ReDiFine is effective robustly.

### E.5 QUANTITATIVE RESULTS

In addition to visual inspections for images generated using ReDiFine, we compare the quantitative results of ReDiFine to baselines with different CFG scales across FID, CLIP score, and Recall. Furthermore, we compute the average sample-wise feature distance (SFD) between a pair of images corresponding to the text prompt as the fidelity metric applicable for text-to-image generation,

$$\text{SFD}_k = \frac{1}{N} \sum d(f(x_{k,i}), f(x_{0,i})) \tag{17}$$

to evaluate each iteration $k$. SFD overcomes the problem of FID being sensitive to the number of images to compare.

DiNOv2 features are used to compute FID, Recall, and SFD. We follow Kynkäänniemi et al. (2019) to compute Recall and set the number of neighbors for computing Recall 5. The results, shown in Figure 33, demonstrate that ReDiFine achieves performance comparable to the optimal CFG scales (2.5 for Pokemon and Kumapi, 1.5 for CelebA-1k and Butterfly) across different datasets and metrics.

## F ABLATION STUDY

This section provides an ablation study to understand how condition drop finetuning and CFG scheduling contribute to the success of ReDiFine.

### F.1 CONDITION DROP FINETUNING

We conducted an ablation study to understand how the probability of dropping text embedding during finetuning affects the image quality in the Chain of Diffusion. We examine 0.1, 0.2, and 0.4 as Stable Diffusion is trained using 0.1 or 0.2. For both Pokemon and CelebA-1k datasets, a probability of 0.2 works the best, as shown in Figure 34 and Figure 35, respectively. Interestingly, condition drop finetuning helps to mitigate the color saturation problem, but its effect decreases with a higher probability. For both of these datasets, condition drop finetuning can mitigate image degradation to some degree, but still, there is a large quality degradation that needs to be improved.

### F.2 CFG SCHEDULING

We also evaluated how different CFG scale decreasing strategies impact image degradation in the Chain of Diffusion. We experimented with two different exponential decay rates and compared them with a linear decreasing strategy. Figure 36 demonstrates that CFG scheduling is effective for Pokemon dataset, generating high-quality images comparable to those generated by ReDiFine. However, as shown in Figure 37, it fails to enhance image quality on CelebA-1k dataset. This highlights the necessity of condition drop finetuning for achieving universal improvements in the Chain of Diffusion across various datasets.

## G ANALYSIS

In this section, we present a series of analyses of images generated through the Chain of Diffusion. Specifically, we examine the distribution of latent values and the differences between conditional and unconditional scores. Additionally, we analyze the power spectra of the images using 2D Fourier transforms and explore fingerprints through forensic analysis (Corvi et al., 2023a;b).

### G.1 LATENT ANALYSIS

Figure 38a illustrates how the distribution of latent values evolves across different iterations. The histograms show the final latent vectors before decoding into pixel space, comparing various CFG scales and ReDiFine. For a CFG of 1.0, the latent distribution rapidly converges into a Gaussian-like shape, with its variance shrinking over iterations. This behavior is consistent with previous work (Bertrand et al., 2023; Alemohammad et al., 2023; Dohmatob et al., 2024b), which theoretically predicted that the self-consuming loop progressively trims the tails of the distribution, reducing output diversity until it collapses to a single mode. We hypothesize that this narrowing in the latent space

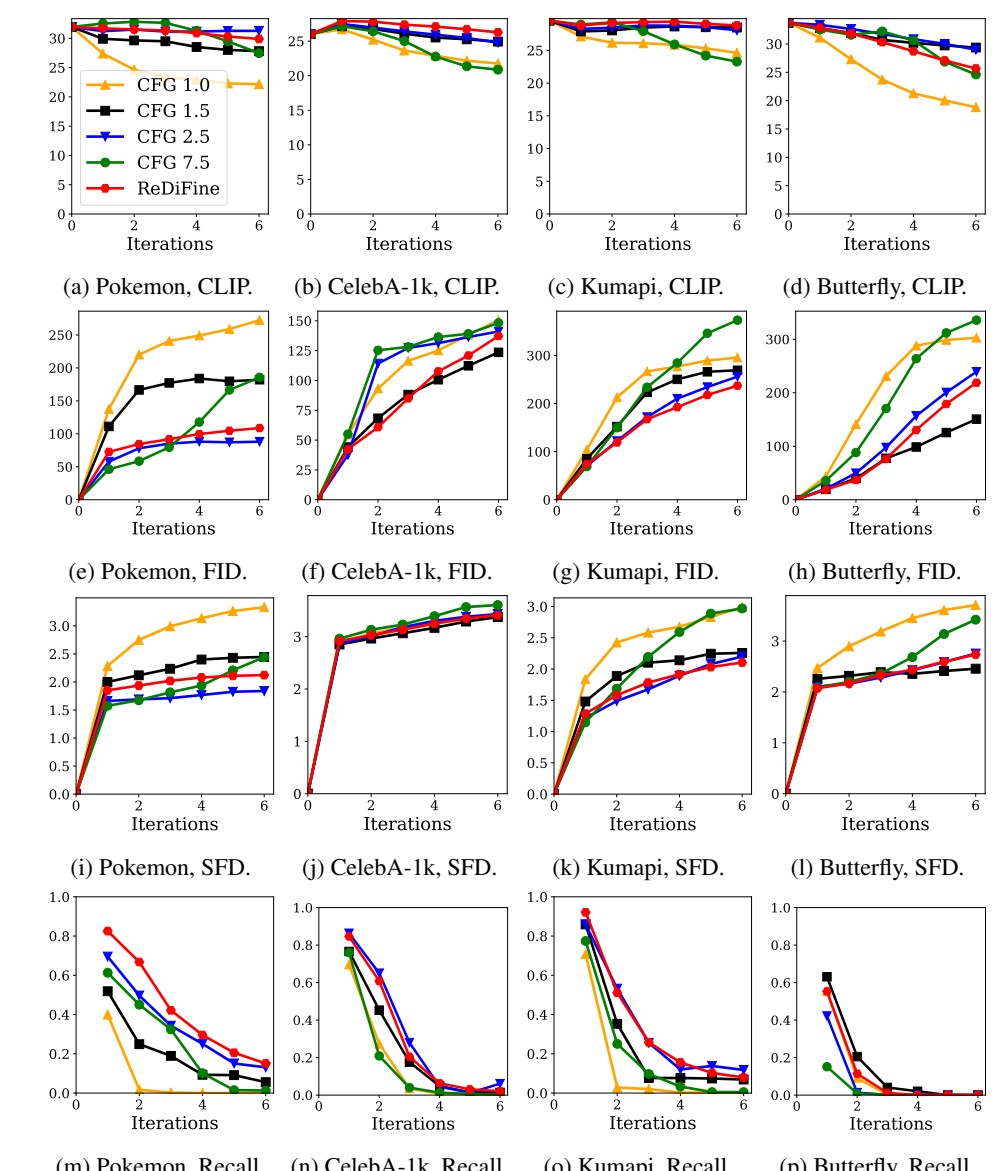

Figure 33: Quantitative results of ReDiFine and baselines (different CFG scales).

leads to blurrier, more homogeneous outputs in pixel space. Conversely, at a CFG scale of 7.5, the latent distribution develops longer tails and tends toward a more uniform spread across space. A CFG scale of 2.5, which demonstrates the best reusability among the three, better preserves the latent distribution over iterations. ReDiFine further enhances this preservation, maintaining the histogram from the first to the last iteration, thus achieving both high fidelity in the first iteration and better reusability.

## G.2 DIFFERENCES BETWEEN CONDITIONAL AND UNCONDITIONAL SCORES

Next, we plot the evolution of the average norm of Diff (= Cond Score − Uncond Score) across diffusion steps for different iterations in Figure 38b. In the first iteration, the highest Diff value is observed for CFG 1.0, followed by CFG 2.5 and CFG 7.5. This behavior can be interpreted as the models' adaptive behavior to preserve the values added to the latent vectors, Diff multiplied by CFG scale, at each step. However, this trend shifts in later iterations. The Diff value for CFG 7.5 continues to grow with each iteration, and by iteration 6, we see elevated Diff values throughout the

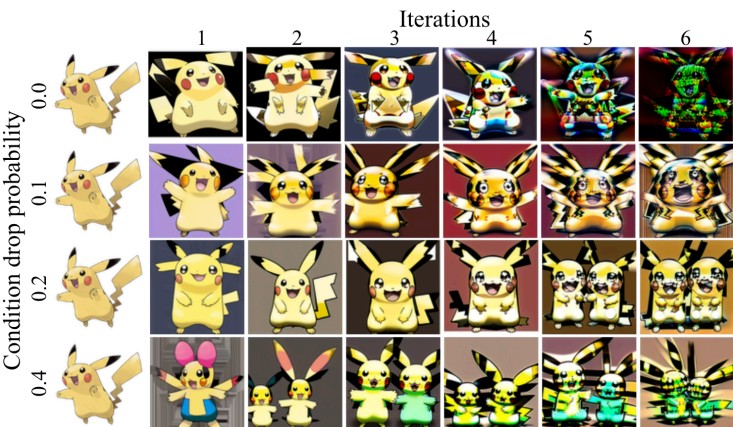

Figure 34: Chain of Diffusion with condition drop finetuning on Pokemon dataset.

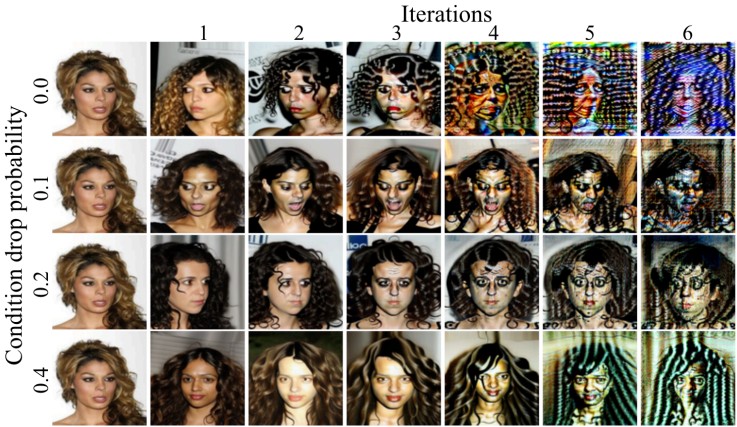

Figure 35: Chain of Diffusion with condition drop finetuning on CelebA-1k dataset.

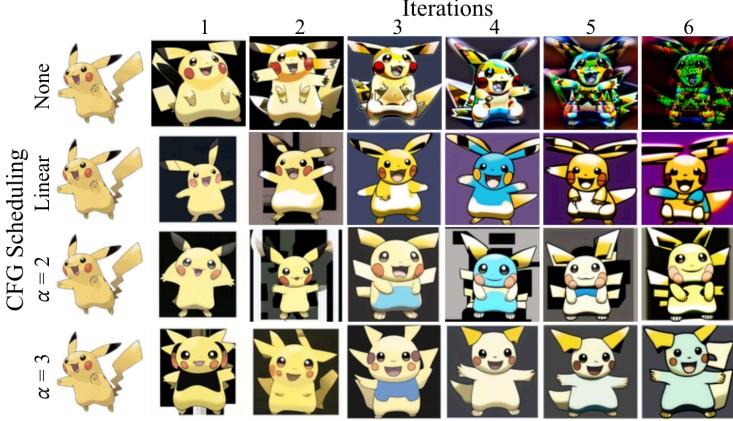

Figure 36: Chain of Diffusion with CFG scheduling on Pokemon dataset.

entire diffusion steps, creating a significant gap compared to CFG 2.5 and 1.0. We conjecture that this accumulation of Diff is the responsible for the high-frequency degradation in images generated with CFG 7.5. In contrast, the Diff value for CFG 1.0 remains relatively stable or even decreases across iterations. The deviation of Diff among different iterations is minimized by ReDiFine, which explains its ability to preserve image quality in later iterations. While condition drop finetuning

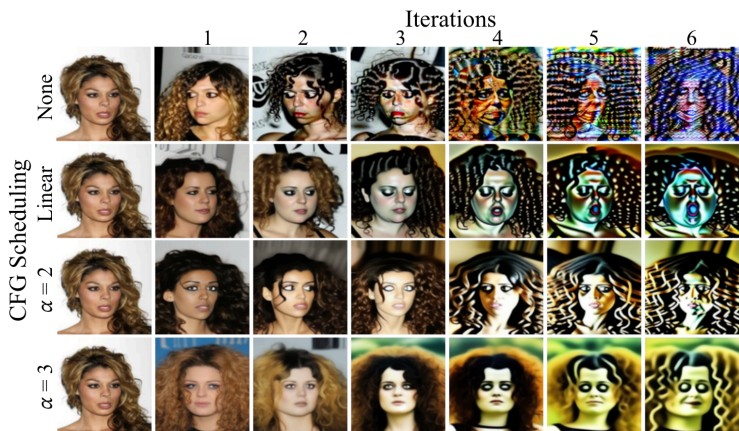

Figure 37: Chain of Diffusion with CFG scheduling on CelebA-1k dataset.

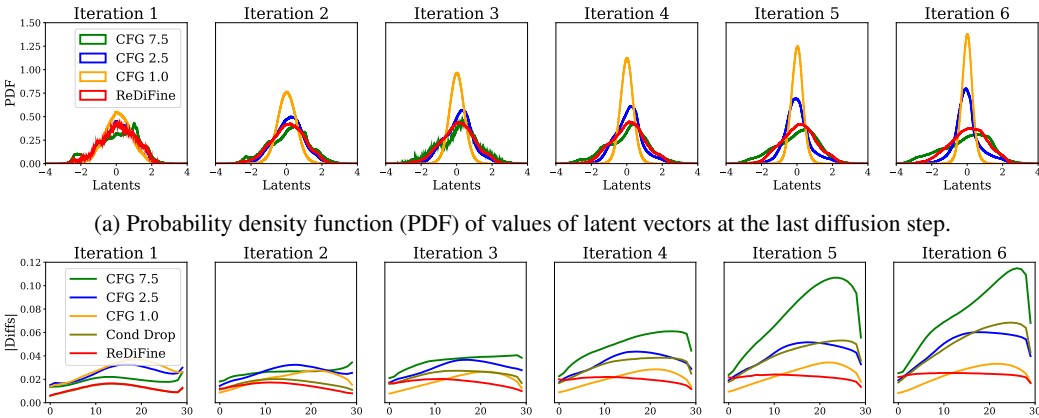

(a) Probability density function (PDF) of values of latent vectors at the last diffusion step.

(b) Norm of differences between conditional and unconditional scores during diffusion steps.

Figure 38: Histogram of latent values and Diffs during diffusion steps for Pokemon dataset. (a) **Latent distribution shrinks over iteration for low CFG and expands with high CFG.** Larger values in latent vectors are more likely to occur with high CFG, gradually increasing the tail of the distribution. (b) **Differences between conditional and unconditional scores increase as the training set is more degraded.** Especially, high differences in the later diffusion steps can be a cause of high-frequency degradation.

helps reduce the Diff in the earlier iterations, it fails to prevent accumulations in later iterations. This limitation is also evident in the ablation study, where condition drop finetuning alone was insufficient to prevent model collapse. Notably, ReDiFine produces significantly smaller Diff values compared to the baseline with CFG scale 2.5, comparable to CFG scale 1.0 even when using a high CFG scale 7.5. This underscores the importance of combining condition drop finetuning with CFG scheduling.

### G.3 POWER SPECTRA OF 2D FOURIER TRANSFORMS

Figure 39 demonstrates the radial and angular spectrum power density of both the original and synthetic images. It is evident that ReDiFine closely maintains the radial spectrum power density of the original training set, whereas even a CFG scale 2.5 falls short. Additionally, ReDiFine demonstrates stable angular spectra throughout the Chain of Diffusion, even though they differ from those of the original training set. Pokemon dataset is used for power spectra analysis.

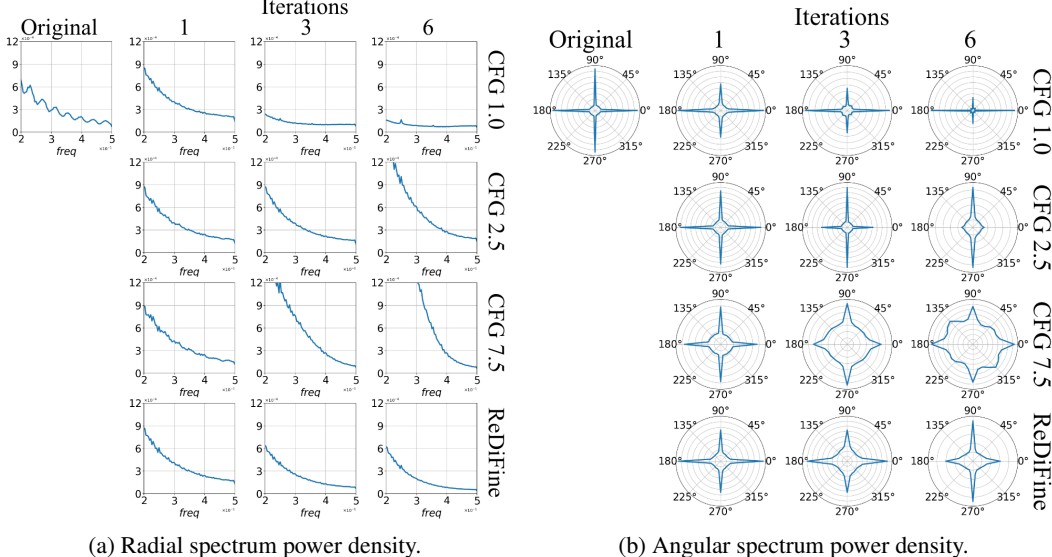

(a) Radial spectrum power density. (b) Angular spectrum power density.

Figure 39: Power spectrum density of the original training set and synthetic sets for Pokemon dataset. **Images generated by ReDiFine maintain power density distribution during Chain of Diffusion while baselines fail. Even CFG scale** 2.5 **cannot maintain the distribution for the last iteration.** (a) Radial spectrum power density. ReDiFine shows a density distribution similar to that of the original training set. (b) Angular spectrum power density. Power density of generated images by ReDiFine remains during the iterations while baselines cannot maintain angular distribution.

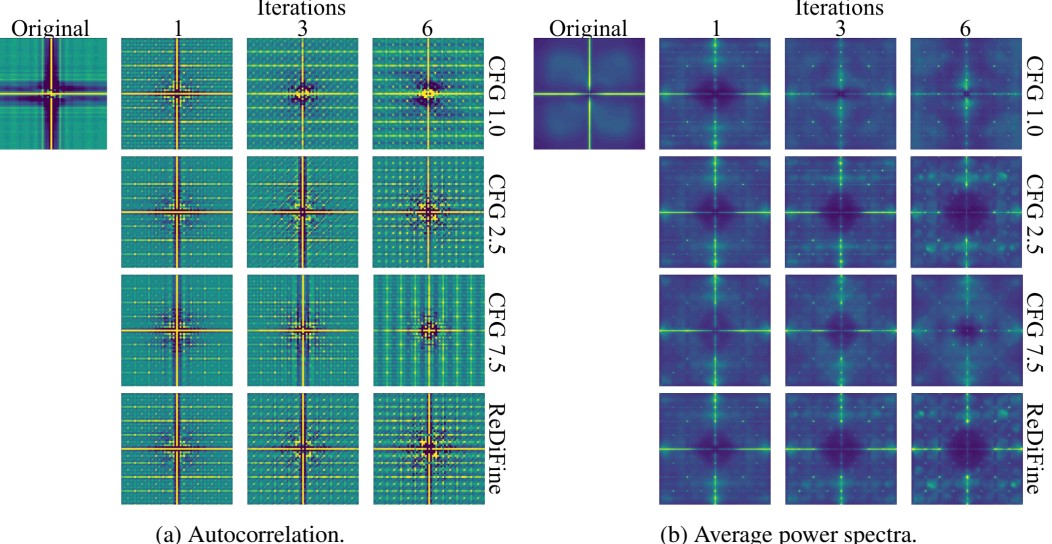

(a) Autocorrelation. (b) Average power spectra.

Figure 40: **The fingerprints of the original training set and synthetic sets show clear differences, and ReDiFine produces fingerprints similar to CFG scale 2.5.** (a) Autocorrelation of image fingerprints. Horizontal and vertical lines gradually disappear for CFG scales 1.0 and 7.5 while they are maintained for CFG scale 2.5 and ReDiFine. (b) Average power spectra of images. Central regions are amplified or diminished for CFG scales 1.0 and 7.5, demonstrating low and high-frequency degradation.

## G.4 FINGERPRINTS FOR FORENSIC ANALYSIS

Several works (Corvi et al., 2023a;b) aim to identify fingerprints of synthetic images. High-quality synthetic images from different generative models have clearly distinct fingerprints, showing the

potential to be used for synthetic image detection. We analyze fingerprints of synthetic images for different CFG scales and iterations, and compare them to fingerprints of the original training set. Both autocorrelation and average power spectra show clear differences between the original training set and synthetic images, as shown in Figure 40. Moreover, how the fingerprints of synthetic images evolve throughout the Chain of Diffusion differ for ReDiFine and different CFG scales. Specifically, fingerprints of synthetic images from ReDiFine are similar to those of images from CFG scale 2.5, while other CFG scales (1.0 and 7.5) make fingerprints different from the first iteration as iterations proceed. Horizontal and vertical lines in autocorrelation gradually disappear and central regions in power spectra vary for further iterations. Also, the varying central regions in power spectra imply that low frequency features increase and decrease for CFG scale 1.0 and 7.5, respectively, aligning with visual inspections. Generating images with fingerprints similar to those of the original real images can be an interesting future direction to reduce the degradation in the Chain of Diffusion. Pokemon dataset is used for fingerprint analysis.

