# OpenReview forum: "Model Collapse in the Chain of Diffusion Finetuning: A Novel Perspective from Quantitative Trait Modeling"
_ICLR.cc/2025/Conference — Submitted to ICLR 2025_

### Official Review · Reviewer_XuJH · 2024-10-30

**Soundness:** 3
**Presentation:** 2
**Contribution:** 3
**Rating:** 6
**Confidence:** 4

**Summary:**

This paper explores the phenomenon of "model collapse" and identifies significant degradation in image quality caused by the iterative process of generation and training. The key factor contributing to this degradation is found to be the CFG. The paper introduces a novel theoretical analysis based on quantitative trait modeling to explain this phenomenon and proposes ReDiFine, a new method to mitigate model collapse without the need for hyperparameter tuning.

**Strengths:**

1. The observation that the direction of degradation changes with different CFG values is intriguing and provides new insights.
2. The proposed ReDeFine method effectively reduces the collapse rate without the need for meticulous tuning. Additionally, this mitigation strategy produces reusable images for future training.
3. The theoretical perspective offers a novel approach to explaining model collapse.

**Weaknesses:**

1. The statement in the abstract that "their outputs are indistinguishable from real data" is too absolute. As noted by [a], the distributions generated by diffusion models still differ significantly from real data from the perspective of classifiers.
2. The experimental setting in this paper differs from existing studies. Here, the authors fine-tune the base model with newly generated data in each iteration, whereas previous studies fine-tune the new model with newly generated data in each iteration. Thus, I am concerned that the analysis presented in this paper may not be directly applicable to prior research.

[a] You Z, Zhang X, Guo H, et al. Are Image Distributions Indistinguishable to Humans Indistinguishable to Classifiers?[J]. arXiv preprint arXiv:2405.18029, 2024.

**Questions:**

1. Different fine-tuning approach:
- Why does this paper fine-tune the base model with newly generated data in each iteration, while existing studies fine-tune the new model in each iteration? I believe this difference makes it distinct from the "model collapse" phenomenon discussed in prior research.
2. Clarification on ‘clip’ metric (line 208):
- What does the 'clip' metric refer to in this context? Could you provide more details?
3. Effectiveness of guidance interval:
- Could the guidance interval proposed by [b] effectively slow down the collapse rate?
4. Alternative fine-tuning strategy:
- Could you try fine-tuning the new model instead of the original model and show some example results?
5. Choice of k=6:
- Why was k=6 chosen for the experiments? Could you elaborate on the reasoning behind this selection?
6. Meaning of symbols in Fig. 2:
- What do the symbols C, W, S, L, and CLIP skip represent in Fig. 2?
7. Fine-tuning the text encoder (Fig. 2):
- Why is fine-tuning the text encoder considered an option in Fig. 2? In my view, the text encoder should not be fine-tuned. Could you explain the rationale for this choice?

[b] Kynkäänniemi T, Aittala M, Karras T, et al. Applying guidance in a limited interval improves sample and distribution quality in diffusion models[J]. arXiv preprint arXiv:2404.07724, 2024.

---

> ### Author Response · Authors · 2024-11-24
> **Rebuttal discussion: Reviewer XuJH**
>
> We appreciate reviewer XuJH's encouragement and rating of 5. We wish this discussion could be an opportunity to raise the rating through comprehensive communication and feedback.
>
> ***
>
> > **[W1] Synthetic image Detection**
>
> This is a great catch! We will revise our phrasing in the final version to “indistinguishable by human eyes” for better clarity.
>
> To comment briefly about synthetic image detection, as you pointed out, detection algorithms have shown to be very effective these days, and filtering images first with a detection tool can be an alternative solution to model collapse. However, we included experimental results showing that **a small fraction of synthetic images slipping through the detection tool can still cause model collapse** (with additional experiments added in the revision for 1-5% synthetic images in **Appendix C.3**). Furthermore, these detection tools still suffer from lower accuracies for out-of-distribution images. So, our approach of reusable image generation can still be a solution that can be used alongside detection tools!
>
> ***
>
> > **[W2, Q1, Q4] Misunderstanding on problem setup**
>
> Thank you for your question! We would like to clarify major misunderstandings about the problem setup of our work. The experimental setting we used aligns with the one proposed by the reviewer: **a new model (pre-trained Stable Diffusion) is trained at each iteration using a set of generated images from the previous iteration**. This setting is illustrated in **Figure 1**, and we will include additional details to clarify this further. Within the same problem formulation as established in prior literature [1, 2], we believe our work contributes significantly by providing extensive empirical results and introducing a novel perspective to better understand model collapse.
>
> [1] Alemohammad, Sina, et al. "Self-consuming generative models go mad." *arXiv preprint arXiv:2307.01850* (2023).
> [2] Guo, Yanzhu, et al. "The curious decline of linguistic diversity: Training language models on synthetic text." *arXiv preprint arXiv:2311.09807* (2023).
>
> ***
>
> > **[Q2] CLIP metric?**
>
> Thank you for this question. We used CLIP to denote the CLIP score for text-to-image alignment:
> $${\rm{CLIP score}} = max(100 * cos(E_{Image}, E_{Text}),0). $$
>
> We will make this clear in the revised version.
>
> ***
>
> > **[Q3] Different guidance mechanism**
>
> Thank you for pointing out this very interesting paper. The idea of applying condition guidance in a limited interval is very intriguing, potentially reduce collapse rate in the Chain of Diffusion. However, it is not evident to us now on how this approach will improve or worsen model collapse. But, benchmarking this work and other newer guidance mechanisms [1, 2] in terms of their model collapse behavior is definitely an interesting and important future direction. We will include a discussion of this point in the final version of the paper.
>
> [1] Tero Karras, Miika Aittala, Tuomas Kynk¨a¨anniemi, Jaakko Lehtinen, Timo Aila, and Samuli Laine. Guiding a diffusion model with a bad version of itself. arXiv preprint arXiv:2406.02507, 2024a.
> [2] Sadat, Seyedmorteza, et al. "CADS: Unleashing the diversity of diffusion models through condition-annealed sampling." ICLR 2024
>
> ***
>
> > **[Q5] Why k=6?**
>
> We restrict the number of iterations to 6 based on our observations that images generated by the 6-th iteration are unrecognizable by human eyes for all datasets. However, we also included experiments up to 12-th iterations in the revised version **Appendix E.2**.
>
> ***
>
> > **[Q6] Prompt symbols and CLIP skip**
>
> The symbols in the table of **Figure 2 (left)** represent Combine, Waifu, Short, and Long, respectively. We have a more detailed explanation in footnote 2. To make this more straightforward, we will include a hyperlink to the footnote for easier reference. CLIP skip is a hyperparameter used in Stable Diffusion, referring to the number of layers to skip (from the output layer) from which the CLIP (text encoder) representation is extracted. We examine this hyperparameter to see if the different layers from the text embeddings contribute to the model collapse.
>
> ***
>
> > **[Q7] Text encoder finetuning?**
>
> Given the severe degradation in image generation, we investigated every possible source of model collapse, including disentangling the impact from UNet and text encoder. For this, we conducted experiments of finetuning just the UNet, just the text encoder, and both. While these choices made some perceptible differences in the generated images, the general trend of model collapse was consistent in all cases.

---

> > ### Comment · Reviewer_XuJH · 2024-11-25
> >
> > Regarding W2: as I mentioned earlier, the experimental setting in this paper is significantly different from previous studies. In the prior works, the new model is fine-tuned with newly generated data in each iteration. In contrast, this paper consistently fine-tunes the base model.
> >
> > Given these differences, I believe all the settings in this paper are fundamentally distinct from those in previous studies. Consequently, the authors have not provided a detailed analysis of prior works, and I think these two approaches cannot be directly compared. Moreover, the paper should revise many of its statements to clarify its position.
> >
> > This is not, as the authors suggested, a misunderstanding.

---

> > > ### Author Response · Authors · 2024-11-25
> > >
> > > Thank you so much for your quick feedback. It seems like we misunderstood your question—we thought you were asking if we are finetuning the model from the previous iteration (i.e., $M_{t-1}$ for the $t$-th iteration), instead of using $M_0$, but in fact, you were saying exactly the opposite! If we understood it correctly this time, your concern is: in previous literature, they finetune the $t$-th iteration model from the new model $M_{t-1}$, whereas we train it from the base model $M_0$, so our setup is not consistent with the literature. First of all, we apologize for our misunderstanding of your comments. We hope our discussion below addresses your concerns.
> > >
> > > ***
> > >
> > > While some papers indeed follow the iterative retraining setting you mentioned [6,7], many notable model collapse papers [1-5] also share the same setting as us!
> > >
> > > - [1]: Even though this is not apparently clear from their Figure 3, in Section 2.1, the authors describe their setting as: “At each generation $t \in \mathbb{N}$, **the model $G^t$ is trained from scratch on the dataset $D^t$**.”
> > > - [2]: Their Figure 1 and 2 are also a bit misleading as it is unclear if $M_2$ is trained from $M_0$ or $M_1$. However, they clarify in Section 5.2: *“We evaluate the most common setting of training a language model – a fine-tuning setting where each of the training cycles **starts from a pre-trained model with recent data**.”* Also, we want to highlight that their theoretical analysis is based on simple statistical abstraction of the process where in the $t$-th iteration the model learns the distribution $p_t$ from the given dataset $D_t$. This does not make any assumption about if the learning starts from $M_{t-1}$ or $M_0$, and the collapse behavior only depends on the evolution of the dataset from $D_0$ to $D_t$. Our abstraction is identical to theirs and thus these analyses are easily applicable regardless of the choice of $M_{t-1}$ or $M_0$.
> > > - [3]: We can see that they also start from a base model in their setup in Figure 1 and they say in the text: *"For the sake of simplicity, **we start from a new instance of the same base model** across different generations, i.e., Base (1) = Base (2) = , ..., = Base (n).”* This makes their setting identical to ours.
> > > - [4]: Their Figure 1 is also unclear if they start from scratch or start from the previous iteration, but they say in the text: *“Subsequently, each **next generation of models is trained from scratch on a new dataset** drawn from the mixed distribution $p_i$.”*
> > > - [5]: Despite their misleading Algorithm 1, their setting is also same as us as they say: “The next generation of generative models **$M_{t+1}$ is then trained from scratch on the new dataset $D_t$.**”
> > >
> > > We want to add a final comment that if we assume that $M_t$ trained/finetuned on $D_{t-1}$ successfully learns the distribution of $D_{t-1}$ at the end, this distinction between whether training $M_t$ starts from $M_0$ or $M_{t-1}$ becomes insignificant. While there can be differences in the difficulty of getting to the optimal $M_t$, in terms of statistical analysis, both cases can share the same principles.
> > >
> > > Hopefully, this clarifies your concern and we will make this point clearer in the related work section in the final paper. But, if we still misunderstood your question, please let us know—we’ll be happy to discuss further! We will also conduct additional experiments to answer your Q4. Thanks again for taking the time to give us feedback :)
> > >
> > > [1] Alemohammad, Sina, et al. "Self-consuming generative models go mad." *arXiv preprint arXiv:2307.01850* (2023).
> > > [2] Shumailov, Ilia, et al. "The curse of recursion: Training on generated data makes models forget." *arXiv preprint arXiv:2305.17493* (2023).
> > > [3] Guo, Yanzhu, et al. "The curious decline of linguistic diversity: Training language models on synthetic text." *arXiv preprint arXiv:2311.09807* (2023).
> > > [4] Fu, Shi, et al. "Towards Theoretical Understandings of Self-Consuming Generative Models." arXiv preprint arXiv:2402.11778 (2024).
> > > [5] Briesch, Martin, Dominik Sobania, and Franz Rothlauf. "Large language models suffer from their own output: An analysis of the self-consuming training loop." *arXiv preprint arXiv:2311.16822* (2023).
> > > [6] Quentin Bertrand, Avishek Joey Bose, Alexandre Duplessis, Marco Jiralerspong, and Gauthier Gidel. On the stability of iterative retraining of generative models on their own data. *arXiv preprint arXiv:2310.00429*, 2023.
> > > [7] Martínez, Gonzalo, et al. "Combining generative artificial intelligence (AI) and the Internet: Heading towards evolution or degradation?." *arXiv preprint arXiv:2303.01255* (2023).

---

> > > > ### Comment · Reviewer_XuJH · 2024-11-26
> > > >
> > > > Thank you for your comments and detailed explanations. Most of my concerns have now been addressed.

---

> > > > > ### Author Response · Authors · 2024-11-27
> > > > >
> > > > > Thank you for your continued engagement. These discussions have been invaluable in helping us consider perspectives we hadn’t previously addressed, and they will strengthen the final version of our paper.
> > > > > We also included the experiment for iterative retraining suggested in Q4 to examine how different CFG scales and ReDiFine perform when one single model is continually finetuned. As demonstrated in Appendix E.4, model collapse consistently occurs for low and high CFG scales under this setting, while the optimal CFG scale ($2.5$ for Pokemon) and ReDiFine can mitigate model collapse. This result demonstrates that (1) model collapse is a universal phenomenon across different problem settings, (2) CFG scale still plays an important role under iterative retraining setting, and (3) ReDiFine is robust to multiple problem settings.
> > > > > We’re glad we could address most of your concerns through our collaborative exchanges. If you feel our updates meet your expectations, we would greatly appreciate it if you could consider adjusting the score to reflect the improvements we've made. Thanks a lot!

---

### Official Review · Reviewer_3oHz · 2024-11-01

**Soundness:** 2
**Presentation:** 3
**Contribution:** 2
**Rating:** 6
**Confidence:** 3

**Summary:**

This  work simulates a recycled pipeline of synthetic images for the training of generative models, and named "self-consuming chain"。
And most importantly, image degradation will happen in such recycle chain. So authors design a metric to evaluate whether this model is collapse.
Authors check and conclude (among many factors) that  higher CFG scale will be the key factor to accelerate the model collapse.
And a new strategy named ReDiFine is proposed to mitigate model collapse, including dropping condition during finetuning and dynamically adjusting the CFG scale along the chain.

**Strengths:**

1. Clear clirify and writing. Good representation.

2. Systematically exploring the self-consuming chain, with designed metric and listed factors.

3. Successfully mitigates the issue proposed by their own.

**Weaknesses:**

1. The methods and the paper is well designed and very serious. And I admire this is a good paper.
But my main concern is the motivation, which seems to be an impractical and pseudo demands:

a. Data hungury now happens in LLM  and many AI models leverage the synthetic data to train. But this work choose the visual/image generation (visual signals are sufficient currently). It might be better to discuss such a topic in the scenario of language?

b. For the training of a large-scale vision model (such as SAM), we have the semi-supervised strategy like human-in-the-loop for SAM.  We can get a huge amounts of raw data with human annotated prompts, for the training of visual generation models.

Combing with the above two reasons, it seems that we do not need using the synthetic data with the same prompts to re-train a visual generation model.

Currently I offer my rating to 5 and I am willing to raise my rating after your replying my main concern.

**Questions:**

1. Could please provide the Recall score in the "self-consuming chain"? A common sense is that higher CFG fator (Within a reasonable range) can lead to better FID but the worse Recall. And recall means the generation diversity.  So I consider that the high CFG is the shallow reason but the worse  and worse generation diversity  brought by high CFG is the key factor in the "self-consuming chain" .

Besides, I recommend you the high-score work CADS[1] in the last ICLR, which is a more elegant CFG to improve both the FID and the Recall score. You can leverage such an idea to sovle the collapse problem in  ReDiFine.

[1] Sadat, Seyedmorteza, et al. "CADS: Unleashing the diversity of diffusion models through condition-annealed sampling." ICLR 2024

---

> ### Author Response · Authors · 2024-11-24
> **Rebuttal discussion: Reviewer 3oHz (Part 1)**
>
> Many thanks for your thoughtful comments, reviewer 3oHz. We are grateful to have had the opportunity to communicate and discuss with you to improve our work.
>
> ***
>
> > **[W1] Motivation for mitigating model collapse in text-to-image models**
>
> This is a fantastic question and I think we can convince you that our work has great practical merits!
>
> We agree that there is a clear challenge of data scarcity in LLM training as models become more and more data-hungry. Also, the comment about semi-supervised learning setting with sufficient amount of human-annotated text-image pairs is interesting. However, we believe that both scenarios reflect a more “large corporate” viewpoint of model training. **The problem setting we are considering takes a “citizens’ viewpoint” of model training, especially artists!** Artists already depend on AI tools and they commonly finetune pretrained model to generate specific styles of art. We see hundreds of new finetuned models released everyday on artist communities like [civitai.com](http://civitai.com/), and observe that finetuning diffusion models (e.g., Stable Diffusion) on small, custom datasets has become a common practice in citizens’ use of AI. Our problem setting is motivated by these observations we made in the artist community.
>
> In this finetuning scenario, data scarcity comes from the fact that the number of images in a specific artist’s style is often limited. Moreover, we would argue that a bigger concern of model collapse is the widespread contamination of Internet by synthetic images. Even if a user wants to filter synthetic images using some detection tools, their accuracy is not 100%, and for out-of-distribution models it is well below 90%. In our work, we show that even 10% of synthetic data in the training set can cause severe model collapse in 6 iterations. This implies that a user can mistakenly include ~10% of synthetic images in their finetuning pipeline and release their generated images on the web, which can be reused by other artists later, creating a self-consuming cycle.
>
> Even for pokemon-style images, we observed that the pre-trained Stable Diffusion model is not able to generate them without finetuning. **We believe that downloading a pretrained model and finetuning for specific artistic style with a small dataset will continue to be a popular adoption of text-to-image models.** While our scenario differs from typical pretraining setting with large-scale datasets, we believe that our work addresses a significant real-world issue.

---

> ### Author Response · Authors · 2024-11-24
> **Rebuttal discussion: Reviewer 3oHz (Part 2)**
>
> > **[Q1] Diversity**
>
> Thank you for raising this excellent point. Diversity (measured as Recall) does decrease in all of our experiments including our proposed method ReDiFine, and this is consistent with previous literature [1, 2, 3]. To address your question, we measured diversity using the Recall metric and included the results in the revised manuscript **(Appendix E.4, Figure 32)**.
>
> Different from the common sense that low CFG scale would increase the diversity, we observe similar patterns to collapse rate that Recall decreases rapidly for both low and high CFG scales. With an optimal CFG scale and ReDiFine, Recall decreases much slowly. **Based on these observations, we agree with the reviewer’s opinion that lower diversity is a more direct factor causing degradations.** However, we want to emphasize again that one of our main contributions is characterizing two distinct directions of degradations and providing a novel perspective to understand them.
>
> [1] Alemohammad, Sina, et al. "Self-consuming generative models go mad." *arXiv preprint arXiv:2307.01850* (2023).
> [2] Quentin Bertrand, Avishek Joey Bose, Alexandre Duplessis, Marco Jiralerspong, and Gauthier Gidel. On the stability of iterative retraining of generative models on their own data. *arXiv preprint arXiv:2310.00429*, 2023.
> [3] Ilia Shumailov, Zakhar Shumaylov, Yiren Zhao, Nicolas Papernot, Ross Anderson, and Yarin Gal. AI models collapse when trained on recursively generated data. *Nature*, 631(8022):755–759, 2024.
> [4] Kynkäänniemi, Tuomas, et al. "Improved precision and recall metric for assessing generative models." *Advances in neural information processing systems* 32 (2019).
>
> ***
>
> > **[Q2] Using CADS for model collapse mitigation**
>
> Thank you for pointing out this very interesting paper. The idea of adding noise in the conditional guidance suggested in paper [1] is very intriguing, and we do agree that this can potentially improve model collapse as this method can preserve diversity better. It is not evident to us on how much this approach will improve model collapse. But, benchmarking this work and other newer guidance mechanisms suggested by other reviewers [2, 3] in terms of their model collapse behavior is definitely an interesting and important future direction. We will include a discussion of this point in the final version of the paper.
>
> [1] Sadat, Seyedmorteza, et al. "CADS: Unleashing the diversity of diffusion models through condition-annealed sampling." *arXiv preprint arXiv:2310.17347* (2023).
> [2] Tero Karras, Miika Aittala, Tuomas Kynk¨a¨anniemi, Jaakko Lehtinen, Timo Aila, and Samuli Laine. Guiding a diffusion model with a bad version of itself. arXiv preprint arXiv:2406.02507, 2024a.
> [3] Kynkäänniemi T, Aittala M, Karras T, et al. Applying guidance in a limited interval improves sample and distribution quality in diffusion models[J]. arXiv preprint arXiv:2404.07724, 2024.

---

> > ### Comment · Reviewer_3oHz · 2024-11-25
> >
> > Many thanks for the replyments and discussions.
> > My main concerns have been addressed.
> > I am willing to raise my score to "6".
> > Looking forward to your future works.

---

> > > ### Author Response · Authors · 2024-11-25
> > > **Many thanks for reading our rebuttal**
> > >
> > > We appreciate your response, and we remain available to answer any additional concerns or questions coming up!

---

### Official Review · Reviewer_wgmS · 2024-11-01

**Soundness:** 1
**Presentation:** 3
**Contribution:** 2
**Rating:** 5
**Confidence:** 3

**Summary:**

The paper investigates an emerging challenge in machine learning: the potential contamination of training data for future models by AI-generated content. The authors introduce and analyze the concept of "model collapse" - a degradation in model performance caused by training on self-generated data. Their key contributions are identifying classifier-free guidance as a primary factor in this degradation process, developing an analytical framework inspired by genetic biology to study this phenomenon, and proposing ReDiFine, a new inference method to mitigate model collapse.

**Strengths:**

- The paper addresses a timely and critical research question, particularly relevant given the rapid proliferation of foundation models and AI-generated content.
- The work demonstrates foresight in identifying and analyzing a challenge that could significantly impact future model development.
- The authors provide clear explanations supported by effective visual diagrams.
- The biological inspiration for the analytical framework offers a novel perspective on the problem.

**Weaknesses:**

- The analysis of the observed phenomena lacks sufficient depth and causal investigation. For instance, the relationship between low CFG scales and low-frequency degradation could be explained by the absence of concept generation in subsequent iterations (those trained without real data), as a CFG scale of 1.0 effectively neutralizes the guidance. However, this hypothesis cannot be verified without samples of generated images used as dataset for successive training iterations.
In this connection, not having the concept in the training images of successive iterations could lead to dilution of the concept getting trained.
- Conversely, the link between high CFG scales and high-frequency degradation could be seen as just concept overfitting. Due to high guidance, crude copies of the previous training images could compose the next the training dataset. This again cannot be verified since training sets for each iteration are not shown. The paper analysis would benefit from a more systematic investigation of these mechanisms, which should form the foundation for developing more robust solutions to the model collapse problem.

**Questions:**

How does the proposed framework extend to multi-domain training scenarios? While early work need not address all complexities, some preliminary insights into situations where multiple concepts are trained simultaneously would better align with the paper's broader motivation of understanding model training with aggregated domains.

Specifically:
- How do different concepts interact during the collapse process?
- Does the rate of degradation vary across different domains?
- How might ReDiFine's effectiveness vary across different concept types?

---

> ### Author Response · Authors · 2024-11-24
> **Rebuttal discussion: Reviewer wgmS**
>
> Thank you so much for reviewer wgmS's valuable questions and comments. We believe that we sufficiently explain the curious connection between our work and concept generation, and we would love to get your further feedback!
>
> ***
>
> > **[W1] Concept disappearing in low CFG scale?**
>
> This is a very interesting comment that concept disappearing is a plausible explanation for the low-frequency degradation. To address this question, we examine the accumulation of training sets—where the training set at iteration T includes data from all preceding generations and the original set(t=0,..T-1)—in an effort to preserve all concepts in the training set. We included results for these experiments in **Appendix C.12** of the revised version. **We observe that a severe model collapse still occurs in the same pattern (images getting more blurry) even though initial concepts are all present in the training set.** We believe that this is a strong evidence that concept disappearing cannot be the sole source of model collapse, and we need additional explanations beyond concept preservation.
>
> ***
>
> > **[W2] Concept overfitting in high CFG scales.**
>
> This is a fantastic observation that there are repeating patterns in the image in later iterations when we use CFG=7.5. **In fact, our direction to understand model collapse in the view of spatial frequency is inspired from this observation as well!** As it is unclear how to reliably measure the repetitions of concepts (objects and/or features), we used spatial frequency as our proxy to capture this behavior. Our experimental results in **Figure 4(b)** demonstrate that spatial frequency successfully captures the distinct behaviors of model collapse, in both low and high CFG settings.
>
> In terms of concept overfitting as a memorization of a concept (understanding a concept dog as an unique species) [1], all model collapse cases can be thought of as some variant of concept overfitting (with distribution shift) as they all have diversity reduction over iterations, regardless of CFG scales (see the Recall plot added in **Figure 32 Appendix E.4** in the revision) .
>
> Overall, we agree that understanding model collapse with the view of concept learning is an interesting approach, but it does not imply the weakness of our angle; **rather, we provide an alternative angle to the problem!** With our unique view of analyzing the problem with spatial frequencies, we were able to get insights theoretically, and also come up with a successful mitigation strategy. We hope we made a clear connection and distinction between concept-based approach and our approach, and we would greatly appreciate it if the reviewer could reconsider the score!
>
> [1] Zeng, Weili, et al. "Infusion: Preventing customized text-to-image diffusion from overfitting." *Proceedings of the 32nd ACM International Conference on Multimedia*. 2024.
>
> ***
>
> > **[Q1, 2, 3] Combining different datasets**
>
> Thank you for the great question! To address the question, we performed an additional set of multi-domain experiments on combining four datasets (Pokemon, CelebA, Kumapi, and Butterfly) to investigate how model collapse affects different domains and concepts. Although the combined dataset includes a larger number of images (~3000) and has more diverse concepts across multiple domains, **severe model collapse still occurs** when the CFG scale is either low (1.0) or high (7.5). Compared to the baseline results for separate datasets **(Appendix B)**, the rate of collapse is only slightly lower—severe degradation begins at iteration 4 in the combined case, whereas it begins at iteration 3 for the baseline—as shown in **Appendix E3** of the revised version. We also observed that ReDiFine successfully mitigates model collapse in the multi-domain setting as well. These additional results highlight that simple approaches of different mixing of data are often insufficient to mitigate model collapse, **but ReDiFine still can be a robust solution in different mixture settings**. Hope this cleared your question and gave you more confidence in our approach! :)

---

> > ### Comment · Reviewer_wgmS · 2024-11-27
> >
> > Thank you for your detailed response to my comments and questions. I appreciate your positive attitude and the effort you put into addressing my concerns.
> >
> > Your inclusion of experiments with cumulative training sets to demonstrate that severe model collapse still occurs despite concept preservation effectively addresses my concern.
> > The discussion of spatial frequency as a proxy for measuring the effects of model collapse, and its relationship to concept overfitting, is well-reasoned.
> > The additional experiments you conducted with a combined multi-domain dataset are much appreciated. The results addressing the interaction of different concepts and the rate of degradation provide a clearer picture.
> >
> > While I appreciate the value of your analysis in advancing our understanding of model collapse with respect to CFG, the mitigation method does not demonstrate sufficient improvement over baseline fixed CFG scales. In particular, the diversity and robustness outcomes remain similar, suggesting limited practical impact. This limits the overall contribution of your work.
> >
> > I have raised my score to 5 to acknowledge the importance of your analysis and the value of the additional experiments addressing my concerns and questions. However, I do not believe the mitigation method is effective enough to substantiate a higher score.
> >
> > Thank you again for your thoughtful response, and I encourage you to continue refining this important line of work.

---

> > > ### Author Response · Authors · 2024-11-30
> > >
> > > We sincerely appreciate your encouraging feedback on our response and are delighted to hear that you recognize the value of our work! As the authors, we would like to emphasize the robustness of the proposed ReDiFine across various datasets and scenarios. It significantly alleviates the need for impractical hyperparameter searches, particularly for CFG scales, which becomes increasingly challenging as the number of iterations increases or the size of the dataset grows.
> > >
> > > We look forward to engaging in constructive discussions during the final stages and remain open to further suggestions and feedback. Thank you once again for your thoughtful and continued engagement!

---

### Official Review · Reviewer_4RPh · 2024-11-04

**Soundness:** 3
**Presentation:** 2
**Contribution:** 3
**Rating:** 6
**Confidence:** 4

**Summary:**

This paper identifies CFG as the major factor leading to model collapses on synthetic data, and analyses it through the perspective of quantitative trait modeling. Then, the paper proposes a novel mitigation strategy: conditional drops during finetuning, and decayed CFG scaling during sampling.

**Strengths:**

* The observations are interesting.
* The analysis of quantitative trait modeling is novel.
* The method is easy to implement and shows promising results.

**Weaknesses:**

* No discussion on the potential tradeoff of the method: specifically, how would the generation quality be affected if using the proposed CFG scheduling?
* Unclear how the method behaves on other CFG scales and decay rates, especially a lower CFG scale, as I only found 5 to 10 in the paper.
* The paper summarizes prior arts as “focused on the reduction of diversity”. The paper could have shown if the proposed method maintains sampling diversity as well.
* The connection between the proposed method and analysis is not convincing. How would a decaying CFG schedule correspond to smoothing truncations?

**Questions:**

* How would the generation quality be affected if using the proposed CFG scheduling?
* How does the method behave on other CFG scales and decay rates?
* Does the proposed method maintain sampling diversity?
* How would a decaying CFG schedule correspond to smoothing truncations?
* Another work [1] sheds light on how CFG increases quality but reduces variation. It would be interesting to see if using their method would alleviate model collapse.
* What if different synthetic images have different CFG scales?

[1] Tero Karras, Miika Aittala, Tuomas Kynk¨a¨anniemi, Jaakko Lehtinen, Timo Aila, and Samuli Laine. Guiding a diffusion model with a bad version of itself. arXiv preprint arXiv:2406.02507, 2024a.

---

> ### Author Response · Authors · 2024-11-24
> **Rebuttal discussion: Reviewer 4RPh (Part 1)**
>
> Thank you very much for reviewer 4RPh's detailed review and meaningful questions. We are hopeful that we’ll be able to address most of your concerns in our response below, and we would love to discuss more if it doesn’t resolve your questions!
>
> ***
>
> > **[W1, Q1] Potential tradeoff of the proposed method?**
>
> We provide the trade-off between the first iteration FID and collapse rate (reusability) in **Figure 6**, showing that ReDiFine (orange hexagon) achieves a desirable trade-off (bottom left) in all four datasets. While high CFG scales achieve slightly better first iteration FID, the difference is negligible, and ReDiFine generates both high-quality and reusable images. Moreover, **Figure 32** showcases additional metrics (FID, CLIP score, Recall, and SFD), and ReDiFine achieves favorable performance in all aspects.
>
> ***
>
> > **[W2, Q2] ReDiFine’s performance with different initial CFG scales & decay rates**
>
> ReDiFine is proposed as a parameter-tuning-free method for mitigating model collapse. Essentially, ReDiFine has three parameters it can change—CFG scales, CFG scheduling functions, and condition-drop probability—but our goal is to suggest a robust mechanism with default parameters that work well across many datasets. **We show that one set of parameters we chose as default (CFG=7.5, alpha=2, and condition drop=0.2) performed well in the four different datasets we tested.**
>
> We show additional experiments on an initial CFG scale of 5.0 and 10.0 to show that ReDiFine works robustly in this range **(Figure 26-29)**. Naturally, CFG scheduling with exponential decay wouldn’t work well with a small initial CFG scale as our intuition is that we want to use higher CFG in the beginning to learn high-level concepts from the text but not overfit to these in later diffusion steps. While our default setting is robust across many datasets, if a user wants to tune the initial CFG scale, we recommend it be between CFG 5.0 and 10.0.
>
> We provide a more detailed ablation study in **Appendix F (Figures 33-36)** where either condition-drop finetuning or CFG scheduling is applied independently, with different condition-drop probability or CFG scheduling functions. Interestingly, the effects of these two strategies vary across datasets. For instance, while CFG scheduling alone produces high-quality images for Pokemon, it fails to do so for CelebA dataset, leading to high-frequency degradation. These observations show that combining these two ideas is crucial to be robust across multiple domains, without requiring dataset-specific search for the configurations of CFG scales, scheduling functions, and condition-drop probabilities. As such, we did not investigate the effects of varying combinations of these three parameters. We hope this clarifies our approach and its scope!
>
> ***
>
> > **[W3, Q3] Diversity metric**
>
> Thank you for raising this excellent point. Diversity (measured as Recall) does decrease in all of our experiments, including our proposed method ReDiFine, and this is consistent with previous literature [1, 2, 3]. However, **what ReDiFine achieves is slowing down this collapse substantially and making sure that the collapse direction is not too far off from the original distribution.**
>
> To address your question, we measured diversity using Recall metric and included the results in the revised manuscript, shown in **Figure 32**.  We observe that Recall decreases rapidly for CFG 1.0 and CFG 7.5. With an optimal CFG scale, it decreases much slowly and ReDiFine’s trend is similar to the optimal CFG. And, we want to emphasize again that ReDiFine achieves such slowdown in model collapse without any hyperparameter tuning! In the original manuscript, we mainly considered FID as a figure of merit, which incorporates both quality and diversity into one metric. These additional results of Recall show that diversity also improves with ReDiFine as compared to the baseline of CFG 7.5, which is a valuable result to include in the final version. Thank you again for your suggestion!
>
> For our Recall experiment, we followed the method in [4] with DiNOv2 as the feature extractor and setting $k = 5$ for k-NN estimation.
>
> [1] Alemohammad, Sina, et al. "Self-consuming generative models go mad." *arXiv preprint arXiv:2307.01850* (2023).
> [2] Quentin Bertrand, Avishek Joey Bose, Alexandre Duplessis, Marco Jiralerspong, and Gauthier Gidel. On the stability of iterative retraining of generative models on their own data. *arXiv preprint arXiv:2310.00429*, 2023.
> [3] Ilia Shumailov, Zakhar Shumaylov, Yiren Zhao, Nicolas Papernot, Ross Anderson, and Yarin Gal. AI models collapse when trained on recursively generated data. *Nature*, 631(8022):755–759, 2024.
> [4] Kynkäänniemi, Tuomas, et al. "Improved precision and recall metric for assessing generative models." *Advances in neural information processing systems* 32 (2019).

---

> ### Author Response · Authors · 2024-11-24
> **Rebuttal discussion: Reviewer 4RPh (Part 2)**
>
> > **[W4, Q4] Connection between ReDiFine and mutation**
>
> We draw a parallel between diffusion generation with constant CFG scales and the two-sided truncation illustrated in **Figure 4(a)**, where different CFG scales correspond to varying truncation ratios. While constant CFG scales enforce a strict selection (rectified cut-off) within the distribution, we hypothesize that dynamic CFG scheduling during diffusion steps results in a more natural selection distribution resembling soft cut-off with exponential distribution, as shown in **Appendix D.2.3**. We further corroborate this hypothesis by running quantitative trait simulation with such soft cut-off selection, and this matches ReDiFine results closely **(Figure 25 in Appendix D.2.3)**. This however is a high-level conceptual connection, and exploring the deeper connections between model collapse and genetic biology is an intriguing avenue for future research.
>
> ***
>
> > **[Q5] Another paper about guidance**
>
> Thank you for pointing out this very interesting paper. The idea of using a weak model as a conditional guidance in this paper is very intriguing. It is not evident to us on how this approach will improve or worsen model collapse. But, benchmarking this work and other newer guidance mechanisms [1, 2] in terms of their model collapse behavior is definitely an interesting and important future direction. We will include a discussion of this point in the final version of the paper.
>
> [1] Sadat, Seyedmorteza, et al. "CADS: Unleashing the diversity of diffusion models through condition-annealed sampling." ICLR 2024.
> [2] Kynkäänniemi T, Aittala M, Karras T, et al. Applying guidance in a limited interval improves sample and distribution quality in diffusion models[J]. arXiv preprint arXiv:2404.07724, 2024.
>
> ***
>
> > **[Q6] Prompt-individual CFG**
>
> That is a great question! We also think that having a dataset composed of images generated with different CFG scales might be more realistic as a user might simply scrape the internet. However, it is impossible to investigate all possible mixtures, so instead, we control the problem framework to have a set of images generated from an unified setting (one CFG scale) to effectively disentangle the effects of various factors. We conjecture that the model collapse behavior of a mixture will resemble interpolation between the behaviors of different CFG scales, but this would not necessarily mitigate the model collapse problem entirely.

---

> > ### Comment · Reviewer_4RPh · 2024-11-25
> >
> > Thanks for your comments, and my concerns are mostly addressed.

---

> > > ### Author Response · Authors · 2024-11-26
> > >
> > > Thank you for your continued feedback!
> > > We are glad to hear that we were able to address most of your concerns.
> > > It was a fruitful time to discuss our work with you.
> > > If you feel our updates meet your expectations and are meaningful, we would greatly appreciate it if you could consider adjusting the score to reflect the improvements we've made :)

---

### Author Response · Authors · 2024-11-24
**General responses to all reviewers**

Dear Reviewers,

We appreciate you taking the time to provide valuable reviews of our work. We are pleased to hear both positive and negative feedback, and we hope our responses answer all the concerns and questions you suggested in the first review.

If our response does not provide satisfactory answers or explanations for your review, please let us know, and we will be glad to communicate and discuss it with you to resolve any misunderstandings between us.

In order to reduce any possible confusion during the rebuttal period, we have not revised the main script. Instead, additional experiment results can be found in the Appendix. Specifically,
- We revised Figure 8-11 in Appendix B by adding one more row for visual inspection for better understanding.
- We revised Figure 14 and 15 in Appendix C.3 by adding additional rows indicating more mixing ratios.
- We included Appendix C.12 for iteration accumulation experiments.
- We included Appendix E.2 for experiments with additional iterations.
- We included Appendix E.3 for dataset accumulation experiments.
- We revised Figure 32 in Appendix E.4 by adding plots for Recall as a measure of diversity.

Please refer to these sections of the revised version for additional experiments mentioned in the rebuttal discussion.

Again, thank you for your valuable time in reviewing our work, and we hope to have further discussion about our responses.

---

### Meta-Review · Area_Chair_VE73 · 2024-12-23

**Metareview:**

This paper investigates the issue of contamination of training data with generated content in the context of text-to-image diffusion models. The paper identified that classifier-free guidance is a primary source of degradation. The paper also proposes an analytical framework inspired by genetic biology to study this phenomenon. Finally, a method called ReDiFine is designed to mitigate the issue, which includes dropping condition during finetuning and dynamically adjusting the CFG scale along the chain.

**Additional Comments On Reviewer Discussion:**

During rebuttal, the reviewers raised concerns about discussions of trade-offs, missing depth of analysis, suitability of problem statement for LLMs, etc. Authors attempted at addressing these concerns.

As mentioned by reviewer wgmS, the mitigation method does not demonstrate sufficient improvement over baseline fixed CFG scales. In particular, the diversity and robustness outcomes remain similar, suggesting limited practical impact. I agree with the reviewer on this.

I feel this paper has some interesting observations about classified-free guidance being the source of degregation in quality when using synthetic data, But, analysis on showing this observation was a bit lacking. I feel the paper can improve if more concrete analysis can be done, and with more advanced image generation models such as Flux. I feel at this moment, the paper is not ready for publication.

---

### Decision · Program_Chairs · 2025-01-22

Reject